# Multi-isotopes in human hair: A tool to initiate cross-border collaboration in international cold-cases

Clément P. Bataille[1]*, Saskia T. M. Ammer[2,3], Shelina Bhuiyan[1], Michelle M. G. Chartrand[1¤], Gilles St-Jean[1], Gabriel J. Bowen[4]

**1** Department of Earth and Environmental Sciences, University of Ottawa, Ottawa, ON, Canada, **2** Geology & Geochemistry Cluster, Vrije Universiteit Amsterdam, Amsterdam, The Netherlands, **3** Co van Ledden Hulsebosch Centre (CLHC), Amsterdam, The Netherlands, **4** Department of Geology & Geophysics and Global Change & Sustainability Center, University of Utah, Salt Lake City, UT, United States of America

¤ Current address: National Research Council Canada, Government of Canada, Ottawa, ON, Canada
* cbataill@uottawa.ca

**Data Availability Statement:** All relevant data are within the paper and its Supporting Information files.

## Abstract

Unidentified human remains have historically been investigated nationally by law enforcement authorities. However, this approach is outdated in a globalized world with rapid transportation means, where humans easily move long distances across borders. Cross-border cooperation in solving cold-cases is rare due to political, administrative or technical challenges. It is fundamental to develop new tools to provide rapid and cost-effective leads for international cooperation. In this work, we demonstrate that isotopic measurements are effective screening tools to help identify cold-cases with potential international ramifications. We first complete existing databases of hydrogen and sulfur isotopes in human hair from residents across North America by compiling or analyzing hair from Canada, the United States (US) and Mexico. Using these databases, we develop maps predicting isotope variations in human hair across North America. We demonstrate that both $\delta^2$H and $\delta^{34}$S values of human hair are highly predictable and display strong spatial patterns. Multi-isotope analysis combined with dual $\delta^2$H and $\delta^{34}$S geographic probability maps provide evidence for international travel in two case studies. In the first, we demonstrate that multi-isotope analysis in bulk hair of deceased border crossers found in the US, close to the Mexico-US border, help trace their last place of residence or travel back to specific regions of Mexico. These findings were validated by the subsequent identification of these individuals through the Pima County Office of the Medical Examiner in Tucson, Arizona. In the second case study, we demonstrate that sequential multi-isotope analysis along the hair strands of an unidentified individual found in Canada provides detailed insights into the international mobility of this individual during the last year of life. In both cases, isotope data provide strong leads towards international travel.

**Funding:** This project was supported by the Chemical, Biological, Radiological-Nuclear and Explosives Research and Technology Initiative [Award No. CRTI08-0116RD] awarded to G.S.J., the Canadian Security and Safety Program Targeted Investment [CSSP-2018-TI-2385] awarded to C.P.B. This research was further supported by the American Academy of Forensic Sciences (AAFS) Humanitarian and Human Rights Resource Center (HHRRC) grant, supported by the AAFS and National Institute of Justice (U.S. Department of Justice), awarded to S.T.M.A. Work by G.J.B. was supported by US National Science Foundation grants DBI-1565128 and DBI-1759730.

**Competing interests:** The authors have declared that no competing interests exist.

## Introduction

The establishment of Interpol many decades ago came from the necessity for international police cooperation in response to the rise of international crime organizations. Nevertheless, such international cooperation usually remains restricted to organized crime while significant barriers impede cooperation in the resolution of cold cases [1]. Countries lack a universal language, funding and the political will to harmonize law enforcement practices and facilitate cross-border cold case investigations [1]. A significant and increasing number of cold-cases are international, particularly in areas located close to land borders where remains of migrants or trafficking victims are common [2]. The remains of individuals with a foreign origin often stay unidentified due to the absence of documents, evidence and cooperation between law enforcement agencies [2]. When DNA or fingerprint databases are available (e.g., EU, North America), they strongly favor cross-border collaboration for those cases with potential international origin [3]. However, while DNA is considered the holy grail of identification, it is often inapplicable either because the DNA is too degraded for analysis, too costly or simply not useful because there is no known reference sample (e.g., family reference sample or the decedent's DNA profile in a database) to compare the sample of the deceased individual to. It has been found that the more socially marginalized the family of the missing is, the more obstacles the family has to face to obtain information about the loved one's whereabouts and to submit data to the appropriate authorities [4]. The administrative, technical, and cultural barriers that limit cooperation between law enforcement agencies are unlikely to disappear in the short term. There is therefore an urgent need to develop tools that provide robust leads to facilitate cross-border implications in the resolution of cold-cases [5].

Isotopic measurements are commonly expressed in delta notation:

$$\delta(\text{X, sample}) = (\text{R}_{\text{sample}}/\text{R}_{\text{standard}} - 1)$$

where R is $n$(heavy isotope)/$n$(light isotope) of element X in the sample [6]. Isotope delta measurements reported in this work are relative to the following standards: Vienna Standard Mean Ocean Water-Standard Light Antarctic Precipitation (VSMOW-SLAP) for hydrogen isotope delta ($\delta^2$H) values, Vienna Canyon Diablo Troilite (VCDT) for sulfur isotope delta ($\delta^{34}$S) values, Vienna Peedee Belemnite (VPDB) for carbon isotope delta ($\delta^{13}$C) values, and AIR for nitrogen isotope delta ($\delta^{15}$N) values. The isotope delta values are typically reported in permil (‰), with an extraneous multiplication factor of 1,000 sometimes appearing in the equation [6]. Stable isotopes are ubiquitous intrinsic markers with the potential to contribute new and actionable evidence in cold-cases, even those involving long-term unidentified individuals [7–17]. Isotopes compose all organic molecules and their abundances in human tissues inform about a person's diet, health, or mobility history providing critical information about human remains [7, 10, 16, 18–22]. Isotopic data from hair have been increasingly used in investigating cold-cases because hair is easily collected and resistant [23], and isotope data in hair are usually preserved post-mortem [24, 25]. While bulk isotope data are often used in forensic cases, sequential isotope profiles along human hair can provide chronological information about the diet and location changes of an unidentified decedent at approximately monthly resolutions as hair grows at ~10 mm/month, though not continuously [7, 9, 11, 12, 16, 26–33]. Isotope data in hair could provide a key screening tool to identify those cold-cases that have a potential international outlook. However, the application of this isotope geolocation technology at the international scale requires the development of cross-border databases of isotope composition in hair and models predicting the isotope patterns in the tissue of interest.

Hydrogen, carbon, nitrogen, oxygen and sulfur are assimilated into hair keratin, and are resistant to post—mortem exposure [34, 35], and are thus the ideal candidate for multi-isotope geolocation of humans from their hair. The $\delta^2$H and oxygen isotope delta ($\delta^{18}$O) values in human hair primarily reflects the isotopic variability in drinking water which follows systematic spatiotemporal patterns along latitude, elevation, and continentality [33, 36].When an individual moves through the landscape, their hair incorporates the isotopic composition of the drinking water consumed along the way and can thus be used to reconstruct origin and movements [36]. Though limited in the geographic resolution of the information it can provide, $\delta^2$H measurements in human hair has been used for decades in isotope geolocation studies [e.g., 32] and data are available from many countries to develop continental-scale maps of isotope variations (or isoscapes) [37].

The $\delta^{34}$S value in human hair is primarily integrated from the isotopic composition of the food consumed, with little isotopic fractionation [38]. Plants and crops, at the base of food systems, uptake sulfur from two main sources: sulfur-containing bedrock minerals or inorganic fertilizers, which generally have low (more negative) but variable $\delta^{34}$S (−15 to 15 ‰), and marine aerosols which have high (more positive) $\delta^{34}$S values (>15 ‰) [39]. The mixing of these two isotopically distinct sulfur sources controls a large part of $\delta^{34}$S variations observed in ecosystems and ancient human societies with high $\delta^{34}$S values in coastal environments and progressively lower $\delta^{34}$S moving inland [39–41]. In modern times, however, the supermarket human diet mixes products from multiple distant locations and sources, complicating the interpretation of $\delta^{34}$S values [38, 42–45]. But even in modern societies, the $\delta^{34}$S values in resident human hair display some differences between regions [38, 42–45]. For example, hair of North American residents have distinctively lower $\delta^{34}$S values than those of Asians or Europeans [38, 42–45]. Bataille et al. (2020) further demonstrated that, within a country, systematic and predictable $\delta^{34}$S gradients were present [43]. The predicted $\delta^{34}$S values in hair of Canadian residents could unambiguously distinguish "true" residents from "snowbirds" traveling to the tropics to escape the Canadian winter.

Both $\delta^{13}$C and $\delta^{15}$N values in human hair are resistant to post-mortem exposure [34]. Both carbon and nitrogen come primarily from diet but can, in certain cases, provide geolocation information [46]. $\delta^{13}$C values in human hair primarily track dietary habits of an individual, particularly the proportion of C$_4$ vs. C$_3$ plant-derived products consumed [13, 23, 24, 46]. This is because C$_3$ crops including beet, barley, rice, potato, or wheat have on average more negative $\delta^{13}$C values (~−25‰) [47] than C$_4$ plants including corn, millet, and cane sugar (~−12‰) [47]. However, some $\delta^{13}$C trends, independent of personal dietary choices, also exist at the regional up to the global scales due to the distinct mix of C$_4$ vs. C$_3$ products in food systems [13]. For example, in North America, more C$_3$ crops (e.g., wheat, barley, beats) are cultivated in the north and dry continental interiors whereas more C$_4$ crops (e.g., corn, sugar cane) are cultivated in the south and coastal regions. This spatial organization of food systems leads to distinct $\delta^{13}$C values in human hair transmitted through the preferential consumption of local food [43, 44, 48]. Similar spatial patterns exist for the $\delta^{15}$N in hair at the regional to global scale due to differences in agricultural practises [13]. However, $\delta^{15}$N values in human hair are mostly controlled by dietary choices with more positive $\delta^{15}$N values for individuals eating more seafood in coastal regions [49].

The main objective of this study is to develop a framework to use multi-isotope geolocation from modern human hair to assist in solving international cold cases. We first analyze and/or compile $\delta^2$H and $\delta^{34}$S values from hair of residents of Canada, the United States (US) and Mexico. We use these databases to generate maps predicting $\delta^2$H and $\delta^{34}$S values in hair across North America. We then analyze the $\delta^2$H, $\delta^{34}$S, $\delta^{13}$C, and $\delta^{15}$N values in bulk hair of deceased undocumented border-crossers (UBCs) that died at the Mexico-US border and were later

identified through the Pima County Office of the Medical Examiner. We compare the predicted geographic origins from bulk hair $\delta^2$H and $\delta^{34}$S analysis with their known origin. We finally run a sequential analysis of $\delta^2$H, $\delta^{34}$S, $\delta^{13}$C, and $\delta^{15}$N values along the length of a Canadian cold case. We assess the potential mobility of the individual and the possibility of cross-border traveling.

## Materials and methods

### Ethics statement

The Office of Research Ethics and Integrity of the University of Ottawa approved this research program under protocol number [5–8–19]. Specifically, all sampling and analytical methods used to collect samples and information were in accordance with these regulations. Informed written consent was obtained from all subjects or from their legal guardians in accordance with, and maintained under, IRB regulations.

**Isotopic analysis of residents' hair samples from across North America.** We analyzed or compiled 692 $\delta^{34}$S (S1 Database) and 846 $\delta^2$H (S2 Database) of hair samples from North American residents. Out of these samples, 649 sites have both $\delta^2$H and $\delta^{34}$S data (S3 Database). A key pre-requisite to compare isotope data in hair measured in different laboratories is to ensure that the data are reported on the same isotope delta scale.

To ensure comparability of $\delta^{34}$S between the various data sets compiled in the S1 Database, all results are traceable to the VCDT scale via IAEA-S-1, IAEA-S-2 and IAEA-S-3. The first set of data was obtained from 101 Mexican residents' hair samples collected by Dr. Ammer in 2019 following protocols described in [44]. These samples had been analyzed for $\delta^{34}$S values at UC Davis Stable Isotope Facility in 2018, which used 6 internal reference materials (RMs) (taurine (−2.5 ± 0.2 ‰), hair (2.7 ± 0.2 ‰), whale baleen (17.5 ± 0.2 ‰), mahi-mahi muscle (19.5 ± 0.2 ‰), seaweed (20.8 ± 0.1 ‰) and cysteine (34.2 ± 0.2 ‰)), calibrated against IAEA-S-1, IAEA-S-2 and IAEA-S-3 [50]. The long-term standard deviation for $\delta^{34}$S analyses at UC Davis is 0.4 ‰. The second set of $\delta^{34}$S data was obtained from 60 American residents' hair samples analyzed in Valenzuela et al [38]. The hair samples from this work were analyzed at the University of Utah in 2010 and calibrated to three internal RMs: zinc sulfide (−31.9 ± 0.3 ‰), ground feathers (16.7 ± 0.4 ‰) and silver sulfide (17.9 ± 0.3 ‰), and these internal RMs were calibrated using IAEA-S-1, IAEA-S-2 and IAEA-S-3. However, no USGS42 and USGS43 samples were analyzed with these samples in 2010. As a quality check, USGS42 and USGS43 were subsequently analyzed in 2022 at the University of Utah using the same analytical protocol as [38]. The mean and standard deviation of the measured $\delta^{34}$S values were 7.94 ± 0.06 ‰ (n = 5; USGS42) and 10.37 ± 0.13 ‰ (n = 5; USGS43). The mean values are within the one sigma uncertainty of the certified $\delta^{34}$S values for USGS42 (7.84 ± 0.25 ‰) and USGS43 (10.46 ± 0.22 ‰) [51], which were in turn calibrated against IAEA-S-1, IAEA-S-2 and IAEA-S-3, showing that this dataset is comparable with the first data set. The third set of $\delta^{34}$S data was obtained from 592 Canadian residents' hair samples collected by Dr. Chartrand between 2007 and 2012, and 531 of these samples were analyzed at the Jan Veizer Stable Isotope Laboratory at the University of Ottawa for $\delta^{34}$S values [43]. Prior to analysis, hair was first washed in a series of three baths of 2:1 solution of chloroform:methanol ($CHCl_3$:MeOH), then dried, ground to a powder using a Retsch ball mill, and stored in glass vials until analyzed. RMs and samples were weighed into tin capsules, and analyzed for $\delta^{34}$S using an Elementar Isotope Cube Elemental Analyser (Elementar, Germany) with a Conflow IV (Thermo, Germany) interfaced to the Delta$^+$XP IRMS equipped with a special 6 collector sulfur cups array (SO-$SO_2$ (Thermo, Germany). The EA method was optimized for $SO_2$: both $N_2$ and $CO_2$ were unretained, and the $SO_2$ was trapped and subsequently released to the IRMS. RMs used for

calibration were IAEA-S-1 (−0.3 ‰), IAEA-S-2 (22.7 ‰) and IAEA-S-3 (−32.6 ‰). The values used for IAEA-S-2 and IAEA-S-3 were not the same as used in the other laboratories, however, these values are within the stated uncertainty of these RMs [50]. Analytical precision, based on the reproducibility of the USGS hair standards, is better than ± 0.3 ‰. The mean and standard deviation of the measured $\delta^{34}$S values were 7.58 ± 0.13 ‰ (n = 3; USGS42) and 10.22 ± 0.15 ‰ (n = 3; USGS43). Those values overlapped with the certified $\delta^{34}$S values and uncertainties for USGS42 and USGS43 [51]. Therefore, all three datasets were deemed to be comparable with each other with respect to $\delta^{34}$S measurements.

The $\delta^2$H values from the 846 hair samples in the S2 Database are traceable to the VSMOW-SLAP scale. The $\delta^2$H values were compiled from three datasets. The hair samples from Mexico [44] were analysed for $\delta^2$H values at the Jan Veizer Stable Isotope Laboratory. The $\delta^2$H of the non-exchangeable hydrogen of hair was determined using the comparative analysis approach described by Wassenaar and Hobson [52]. We performed hydrogen isotopic measurements on $H_2$ gas derived from high-temperature (1400˚C) flash pyrolysis (TCEA, Thermo, Germany) of 150 ± 10 µg hair subsamples and keratin standards Caribou Hoof Standard (CBS; −157.0 ± 0.9 ‰), Kudo Horn Standard (KHS; −35.3 ± 1.1 ‰) [53], USGS42 hair (−72.9 ± 2.2 ‰) and USGS43 hair (−44.0 ± 2.0 ‰) [54] loaded into silver capsules. The resultant separated $H_2$ flowed to a Conflow IV (Thermo, Germany) interfaced to a Delta V Plus IRMS (Thermo, Germany) for $\delta^2$H analysis. The hair samples were calibrated to three reference materials: CBS, KHS and USGS43, while USGS42 was was used as a quality check. The measured values for USGS42 (−73.3 ± 0.8, N = 4) were within the reported value and uncertainty, thus verifying this approach. Analytical precision of these measurements is based on the reproducibility of USGS42, and is better than ± 2 ‰.

The $\delta^2$H data from Ehleringer et al. (2008) [33], and 535 Canadian hair samples ($\delta^2$H values measured in 2013, results available in a non-peer-reviewed report [55]), were analyzed using older protocols and calibration standards. To ensure comparability between these two datasets and the Mexican hair measured as described above, we transformed the hair $\delta^2$H values from Ehleringer et al and Chartrand et al using the function *refTrans* in the assignR package to place them on the same calibration scale (calibrated to CBS and KHS), as described in detail in [37]. Although the Mexican hair measured at the Jan Veizer Stable Isotope Laboratory was calibrated using USGS43 in addition to CBS and KHS, the difference in $\delta^2$H values between these calibrations (CBS, KHS, USGS43 vs CBS, KHS) was < 0.7 ‰, and is much less than the combined uncertainties due to measurement and rescaling (~± 3 ‰ [37]).

**Deceased undocumented border crossers.** The remains of deceased undocumented border crossers are often found by US Border Patrol, non-governmental institutions or private citizens in the rural regions along the Mexico-US border. Remains found throughout most of southern Arizona are then transported to the Pima County Office of the Medical Examiner (PCOME). As part of Dr. Ammer's doctoral thesis, hair samples from four deceased undocumented border crossers were collected by pulling the hair with the roots (Table 1). These individuals were found relatively rapidly after their death (<5 weeks) limiting potential effect of decomposition on the isotope values [34]. Additionally, the individuals have been tentatively identified by authorities through various means and await final confirmation. Those identifications can be used, with caution, as a mean to validate isotope-based geographic assignments. All hair samples were approximately 4 cm long, thus representing the last few months of these individuals' lives.

It is assumed that the bulk analysis of isotopes in hair of those UBCs, by and large, reflect the region of last residence. However, the journey to the border can sometimes take up to months and in rare cases, even years as some reside close to the border awaiting their chance to cross. Consequently, the isotope delta values obtained from bulk hair is comprised of a mixture of isotopic signatures of water and food from both the home region and from the

**Table 1. Metadata on the four deceased undocumented border crossers.**

| UBC # | Sex, Age at Death | Country, State, City of Origin | Border Patrol Corridor Found | Surface Management | Latitude/Longitude of recovery | Body condition | Post Mortem Interval |
|---|---|---|---|---|---|---|---|
| UBC #1 | Male, 28 | Mexico, Sinaloa, Jitzamuri | Goldwater | Cabeza Prieta National Wildlife Refuge | 32.3475;-113.3066 (precise to within ca. 300ft./100m) | Decomposed w/ focal skeletonization | < 3 weeks |
| UBC #2 | Male, 61 | Mexico, Tlaxcala, Tlacatecpa | San Miguel | Tohono Oodham Nation | 31.8997;-111.769884 (precise to within ca. 300ft./100m) | Decomposed w/ focal skeletonization | < 3 weeks |
| UBC #3 | Male, 27 | Mexico, Guerrero, Ayulta de los Libres | San Miguel | Tohono Oodham Nation | 31.890533;-112.163068 (precise to within ca. 300ft./100m) | Decomposed w/ focal skeletonization | < 3 weeks |
| UBC #4 | Male, 20 | Guatemala, Huehuetenang, San Ildefonso | San Miguel | Tohono Oodham Nation | 31.735367/-111.835983 (precise to within ca. 300ft./100m) | Skeletonization w/ mummification | < 5 weeks |

travelling locations. Further, some of the individuals might have crossed the border multiple times after living in the US and being deported. Based on the investigative work and contact with the potential family of the deceased, it is thought that UBC #2 lived in the US for a few months and was subsequently deported and died during his subsequent crossing back into the US. Consequently, these mean bulk hair isotope delta values are not necessarily compatible with the individuals' regions of origin, and due to isotopic mixing, may also suggest stays in regions where they have never been. The metadata on the four deceased undocumented border crossers can be found in Table 1.

The bulk hair from the UBCs were prepared and analyzed for $\delta^2$H at the Jan Veizer Stable Isotope Laboratory using the methods described above (Table 2 and S1 Table). The $\delta^{13}$C, $\delta^{15}$N and $\delta^{34}$S of bulk hair were previously analyzed at UC Davis Stable Isotope Facility and the results are available in Saskia Ammer's doctoral thesis [56].

**Canadian cold-case.** In 2008, the RCMP Halifax Detachment sent a clump of dreadlock hair to the University of Ottawa from an unidentified individual recovered on October 8, 2004 near the Halifax city airport, Nova Scotia, Canada (hereafter called Mr. Halifax; RCMP Case File #2004–3757)). Rather than analyzing the bulk hair, a chronological isotopic profile was obtained following a procedure that was developed and tested in a previous study to align, combine and sequence multiple hair [36]. However, the hair of this individual was extremely brittle which made alignment challenging. Consequently, individual locks of hair were used as the basis for alignment, combination and segmentation. The hair was washed as described

**Table 2. Isotope results of the four deceased undocumented border crossers.**

| UBC # | | $\delta^2$H, ‰ | $\delta^{34}$S, ‰ | $\delta^{13}$C, ‰ | $\delta^{15}$N, ‰ |
|---|---|---|---|---|---|
| UBC #1 | | −51.7 | 6.2 | −15.2 | 10.4 |
| UBC #2 | | −57.4 | 3.6 | −15.2 | 8.5 |
| UBC #3 | | −51.6 | 5.0 | −13.6 | 9.5 |
| UBC #4 | | −57.3 | 3.2 | −13.6 | 9.1 |
| Mr. Halifax | mean ± 1sd | −60.4 ± 2.6 | 2.9 ± 0.8 | −19.1 ± 0.5 | 9.1 ± 0.2 |
| | range | −57.2 to −69.3 | 0.9 to 4.2 | −19.9 to −18.0 | 8.6 to 9.4 |
| Canada | mean ± 1sd | −86.3 ± 12.6 | 1.7 ± 1 | −18.5 ± 0.6 | 9.2 ± 0.5 |
| | range | −60.2 to −120.5 | −1.4 to 4.8 | −20.3 to −16.7 | 7.6 to 10.8 |
| US | mean ± 1sd | −69.9 ± 13.2 | 3.4 ± 1.1 | −17.2 ± 0.8 | 8.9 ± 0.4 |
| | range | −37.2 to −106.2 | 0.9 to 5.0 | −21.6 to −14.7 | 6.5 to 9.7 |
| Mexico | mean ± 1sd | −59.2 ± 6 | 4.5 ± 1.1 | −15.6 ± 1 | 9.2 ± 0.6 |
| | range | -42 to −74 | 2.7 to 8.0 | −18.3 to −12.8 | 6.8 to 10.8 |

previously, and a total of twenty-seven 0.5 cm segmented hair samples, each representing about half a month timeframe, were prepared. Due to the difficulty in aligning the dreadlocks, the time uncertainty and averaging represented by the isotope time-series are probably higher than in previous studies causing, in this case, a more "elastic" timeframe than usual [9, 11, 36]. We analyzed each segment for $\delta^2$H and $\delta^{34}$S, as described previously (S2 Table). $\delta^{13}$C and $\delta^{15}$N were previously analyzed, and the results are available in a non-peer-reviewed report [55].

## Isoscapes

Geographic probability maps compare the observed isotopic value for a biological tissue (e.g., hair) with that predicted by an isoscape [57].

**Sulfur hair isoscape.** Bataille et al. (2020) predicted the $\delta^{34}$S trends of modern human hair across Canada which showed a strong gradient from coast to more inland provinces reflecting the influence of local food systems [43]. Similar trends are observed in the $\delta^{34}$S in hair of residents of the US [38] and, to a lesser extent, Mexico [44]. Based on these observations, we used an existing framework to generate a $\delta^{34}$S isoscape from hair of North American residents [40, 43]. We assembled data on selected covariates that represent the main factors that impact variability in $\delta^{34}$S values, including country of residence (Canada, US or Mexico), geology (age), climate (e.g., precipitation, temperature), soil proprieties (e.g., pH, clay content, organic matter), aerosol deposition (e.g., sea salt) and distance to the coast. We resampled and re-projected all the selected environmental geospatial products into WGS84-Eckert IV 1km$^2$ resolution and used the latitude and longitude of each sampling site to extract the local values for each raster. We combined the $\delta^{34}$S hair compilation and the extracted covariate values at each site into a regression matrix and used generalized linear model (GLM) regression kriging to predict $\delta^{34}$S in hair across North America using the "caret" package [58]. We first use the Akaike information criterion (AIC) to estimate prediction error and rank the relative quality of GLM models. We then used simple kriging to map the remaining variance of the residuals (S1 Script).

**Hydrogen hair isoscape.** We generated a $\delta^2$H isoscape for human hair following the procedure described in the assignR package [59]. The process requires a tissue-specific isotope dataset of known-origin individuals to develop a transfer function between an isoscape and the tissue of interest. We used the compiled dataset of $\delta^2$H in hair combining the $\delta^2$H in hair of Canadian and Mexican residents analyzed in this study with published $\delta^2$H in hair of US residents (rescaled as described previously) and incorporated this dataset to the known origin sample database in the assignR package. We calibrated the hair $\delta^2$H isoscape using the *calRaster* function in the assignR package.

## Probability maps

We used the metrics generated by the *QA* function in assignR to compare the quality of probability maps generated from each isoscape. We then used the continuous-surface probability framework from the assignR package [59] (i.e., *pdRaster* function) to estimate the most likely locations of origin of each individual from the bulk hair single ($\delta^2$H or $\delta^{34}$S) and dual ($\delta^2$H and $\delta^{34}$S) isotope data from deceased undocumented border crossers and from each segment of the Canadian cold-case (S2 Script).

## Results

### Sulfur isoscape

The 692 compiled and analysed $\delta^{34}$S data from hair of North American residents are normally distributed (Shapiro Test, p-value<0.05). $\delta^{34}$S values range from −1.4‰ to 8‰ and average

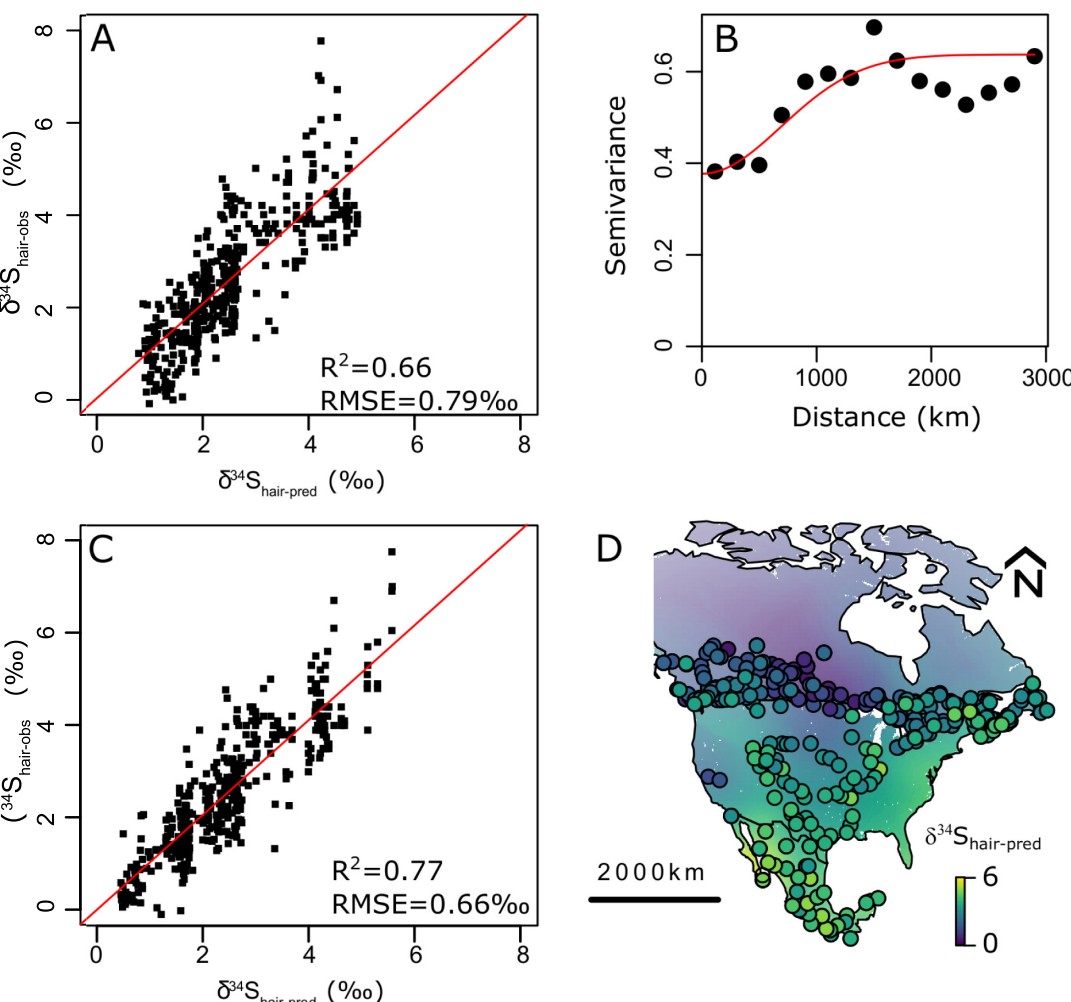

**Fig 1. Sulfur isoscape generated by regression kriging. A:** Cross-validation between measured and predicted $\delta^{34}$S values using a Generalized Linear Model (GLM) regression using distance to the coast, longitude, latitude and country as predictors. RMSE = Root Mean Square Error. The red line is the best-fit linear model. $\delta^{34}$S$_{hair-obs}$ values correspond to the measured $\delta^{34}$S values in human hair **B:** Variogram of the GLM regression residuals. Red line is a Gaussian model fit using simple kriging. **C.** Cross-validation between measured and predicted $\delta^{34}$S values using GLM regression kriging. RMSE = Root Mean Square Error. The red line is the best-fit linear model. $\delta^{34}$S$_{hair-obs}$ values correspond to the measured $\delta^{34}$S values in human hair **D.** Map of $\delta^{34}$S in hair of resident humans across North America with locations of collection sites from residents (including individuals from this study as well as published data) [38, 44]. Coastlines and country boundaries are from http://www.naturalearthdata.com/. The R scripts to generate these figures are available in S1 and S2 Scripts. Data are available in S1 Database.

2.3 ‰. Some geographical regions of North America are underrepresented in the dataset: the eastern US, western US and southern US.

The best GLM model (AIC = 556) used latitude, longitude, country of origin, and distance to the coast as the dominant predictors of the $\delta^{34}$S values (Fig 1A). This GLM accounted for 66% of the variance with a Root Mean Square Error (RMSE) of 0.79‰ (Fig 1A). After kriging the residual values using a Gaussian variogram model (Fig 1B), the resulting regression kriging model accounts for 77% of the variance with a RMSE of 0.65 ‰ (Fig 1C). The value of 0.65‰ represents ~8% of the full range of measured $\delta^{34}$S values over the dataset. Latitude and country of origin are the strongest predictor of $\delta^{34}$S values. Going north to south, Canadian samples, on average, have lower $\delta^{34}$S values than US citizens and Mexicans. Distance to the coast and

longitude are also key predictors with individuals living close to the coast having higher $\delta^{34}$S values than those living more inland.

The regression kriging produced a $\delta^{34}$S isoscape in human hair that displays strong spatial patterns associated with country and distance to the coast. Sites located in coastal North America have higher $\delta^{34}$S values (Fig 1D). The highest values are found in western coastal Mexico. Conversely, sites located in interior regions of Canada have the lowest $\delta^{34}$S values.

## Hydrogen isoscape

The $\delta^2$H data compiled from 846 samples of hair from North American residents are normally distributed (Shapiro Test, p-value<0.05). $\delta^2$H values range from $-119.8$ ‰ to $-37.4$ ‰, with an average of $-79$ ‰. The produced $\delta^2$H isoscape in human hair displays strong spatial patterns (Fig 2A), with a strong correlation between $\delta^2$H values in hair and $\delta^2$H values in precipitation (Fig 2B).

## Isotope-based probability map results

**Quality evaluation of isotope-based probability maps.** Quality metrics for the single- and dual-isotope probabilistic maps suggest strong potential for this method to provide accurate and specific information on the origin of human hair samples, and highlight the added power of the dual-isotope method (Fig 3). The area-exclusion plot (Fig 3A) shows that the spatial distribution of posterior probabilities is very uneven, particularly for the dual-isotope analysis. This implies that the isotopic information strongly discriminates between more- and less-likely regions and could be used to eliminate large parts of North America as a potential origin for a sample at a high level of certainty. The validation plot (Fig 3B) shows that, for hydrogen isotopes, the proportion of samples correctly assigned to origin mostly scales as expected with the probability threshold adopted for assignment, implying that this isoscape and its uncertainty appropriately represent

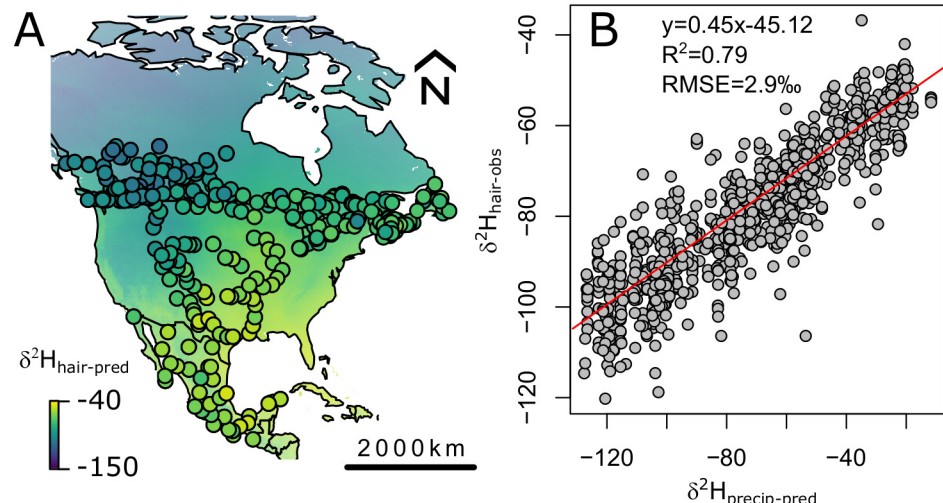

**Fig 2. Hydrogen isoscape generated in the assignR package. A:** Map of $\delta^2$H in hair of residents from North America with locations of collection sites from residents (including individuals from this study as well as published data) [33, 37, 55]. Coastlines and country boundaries are from http://www.naturalearthdata.com/. **B:** Calibration equation between modeled $\delta^2$H in precipitation and hair from North American residents (including individuals from this study as well as published data). The R script to generate these figures is available in S2 Script. Data are available in S2 Database.

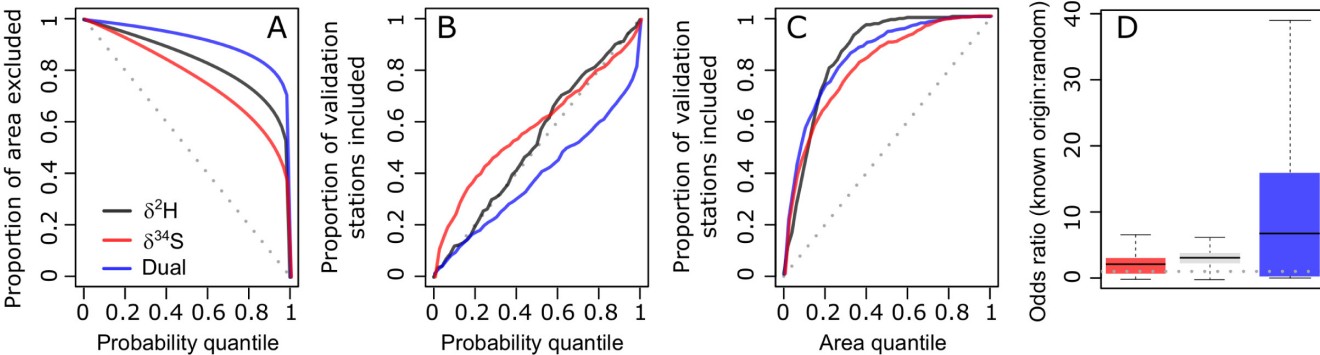

**Fig 3. Quality evaluation of hydrogen, sulfur and dual-isotope probability maps.** Plots were generated using the *QA* function in the assignR package. **A.** Proportion of the study area excluded from the assignments as a function of probability threshold. Higher values indicate a potential for more specific assignments. **B.** Proportion of validation samples correctly assigned as a function of the probability threshold. If accurate posterior probabilities are estimated for each sample, these values should fall along the 1:1 line. **C.** Proportion of validation samples correctly assigned as a function of the area quantile, providing an integrated measure of assignment sensitivity. **D.** The distribution of odds ratios for the known origins of the validation samples relative to random quantifies the strength of isotopic support for one location relative to another. Higher odds ratios indicate more specific assignments. The R script to generate these figures is available in S2 Script.

the human hair data. The validation plots show slight deviation from the expected proportion of samples correctly assigned for $\delta^{34}$S and for dual isotopes (Fig 3B). For $\delta^{34}$S the proportion of stations included are overestimated relative to the probability quantile likely indicating that the modeled uncertainty is lower than represented in the isoscape. Surprisingly, however, the dual isotope show the opposite trend with the proportion of validation stations underestimated relative to the probability quantile indicating that the model misses the true origin of known-origin individuals more than expected (Fig 3B). This could reflect some bias in the sulfur isotope predictions associated with the regression kriging approach and/or the validation approach. This bias would be accounted for in the univariate uncertainty but leads to incorrect predictions in the multivariate case. With larger datasets with more complete spatial coverage, it might be useful to test new modeling approaches to predict $\delta^{34}$S variations (e.g., random forest regression). The assignment power plot (Fig 3C) uses the known origin data to test the ability of the method to correctly assign sample origin across a range of exclusion area thresholds. It shows that both single- and dual-isotope analyses have high power, though the $\delta^{2}$H and dual-isotope method give substantial increase in power across all area thresholds relative to $\delta^{34}$S. Finally, the odds ratios for the known locations of sample origin are substantially higher than the random value for all methods, but are approximately 5 time greater for the dual-isotope method than either single-isotope analysis (Fig 3D). Collectively, these results support the validity of the isoscapes as a template for interpreting human hair isotope data and suggest that assignments made using the isotopic data, particularly in the case of dual-isotope analysis, can provide accurate and specific information on the geographic origin of samples.

**Deceased undocumented border crossers.** The hair samples of the deceased undocumented border crossers were bulk-analyzed, rather than segmented, and represent approximately the last 4 months of life of the individual. Table 2 presents the $\delta^{2}$H, $\delta^{34}$S, $\delta^{13}$C and $\delta^{15}$N isotope values (average of three measurements) for each of the four individuals. The $\delta^{13}$C and $\delta^{15}$N isotope values of the four UBCs were compared to the values previously published for residents of North America [38, 43, 44].

Based on the available $\delta^{15}$N data from North America, an origin from the US can be excluded for UBC #1, and all other individuals fall within the ranges reported for all three countries (Table 2). The $\delta^{13}$C values reported for UBC #3 and UBC #4 would exclude both the

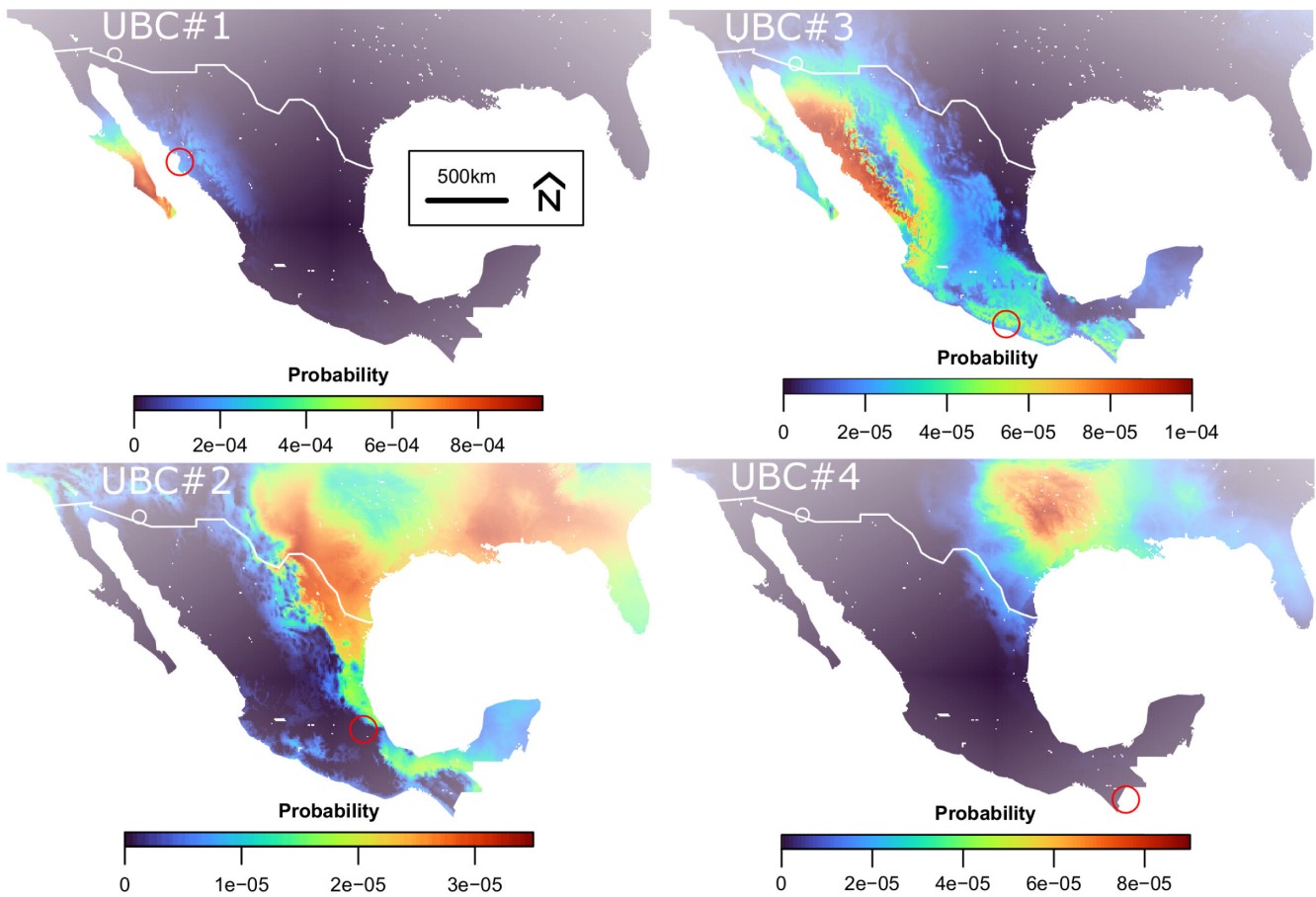

**Fig 4. Probabilistic maps of provenance for deceased undocumented border crossers estimated from $\delta^2$H and $\delta^{34}$S values for bulk hair.** Locations where the human remains were discovered, are indicated by white circles. Locations where the remains are thought to have originated, based in independent evidence, are marked with red circles. Colours on the maps depict the predicted probability of origin based on isotopic evidence. Coastlines and country boundaries are from http://www.naturalearthdata.com/.The R script to generate these figures are available in S2 Script. Data are available in S1 Table and Table 2.

US and Canada as potential countries of origin (Table 2). The $\delta^{13}$C values for UBC #1 and UBC #2 fall within the higher end of the range of values found in the US and Canada but would not allow for a clear distinction. The $\delta^{34}$S values of UBC #2 and UBC #4 fall within the ranges reported for all three countries, while UBC #1 and UBC #3's values are too positive to have originated from Canada (Table 2).

The $\delta^2$H values of all UBCs fall within the ranges reported for Mexico and the US but are not compatible with Canada (Table 2).

We generated single and dual isotopes probabilistic maps of provenance for each individual and compared them to the location of origin tentatively identified by authorities (Fig 4). UBC #1 probably originated from Jitzamuri, Sinaloa, Mexico. Only a few coastal regions of western Mexico are compatible with the isotope data from this individual's hair. Among those, the region around Jitzamuri shows high probability. UBC #2 probably originates from Tlacatecpa, Tlaxcala, Mexico. The dual isotopes probabilistic maps show low probability of origin in the purported region of origin except for some coastal regions along the Gulf of Mexico. Probable regions of origin include most of the southeastern US, and coastal region around the Gulf of Mexico. UBC #3 probably originates from Ayulta de los Libres, Guerrero, Mexico. The dual isotope probabilistic maps show a high probability from this city but are not very specific as all

the western coast of Mexico shows a high probability of origin with highest probability in northwestern Mexico. Most of northern Mexico and the southern US regions are also compatible with the isotope values of this individual. UBC #4 most likely originated from San Ildefonso, Huehuetenango, Guatemala. Unfortunately, our map does not extend to Guatemala. The most probable regions of potential origin within our study areas is the south-central US along the Mississippi river valley.

**Halifax cold-case results.** We compare the average $\delta^{13}C$ (−19.1 ± 0.5 ‰) and $\delta^{15}N$ (9.1 ± 0.3 ‰) values of Mr. Halifax's hair with previously published $\delta^{13}C$ and $\delta^{15}N$ values of North American residents [38, 43, 44] (Table 2). The $\delta^{15}N$ values are similar to the $\delta^{15}N$ values for Canada, the US and Mexico. The profile shows two main zones of stable $\delta^{15}N$ values with values around 9.3 ‰ between 5 and 14 months PTD and a shift to lower values 5 months PTD (Fig 5A). The average hair $\delta^{13}C$ value from Mr. Halifax is most similar to the average bulk hair $\delta^{13}C$ values of Canada's western and central provinces including western Ontario, Manitoba,

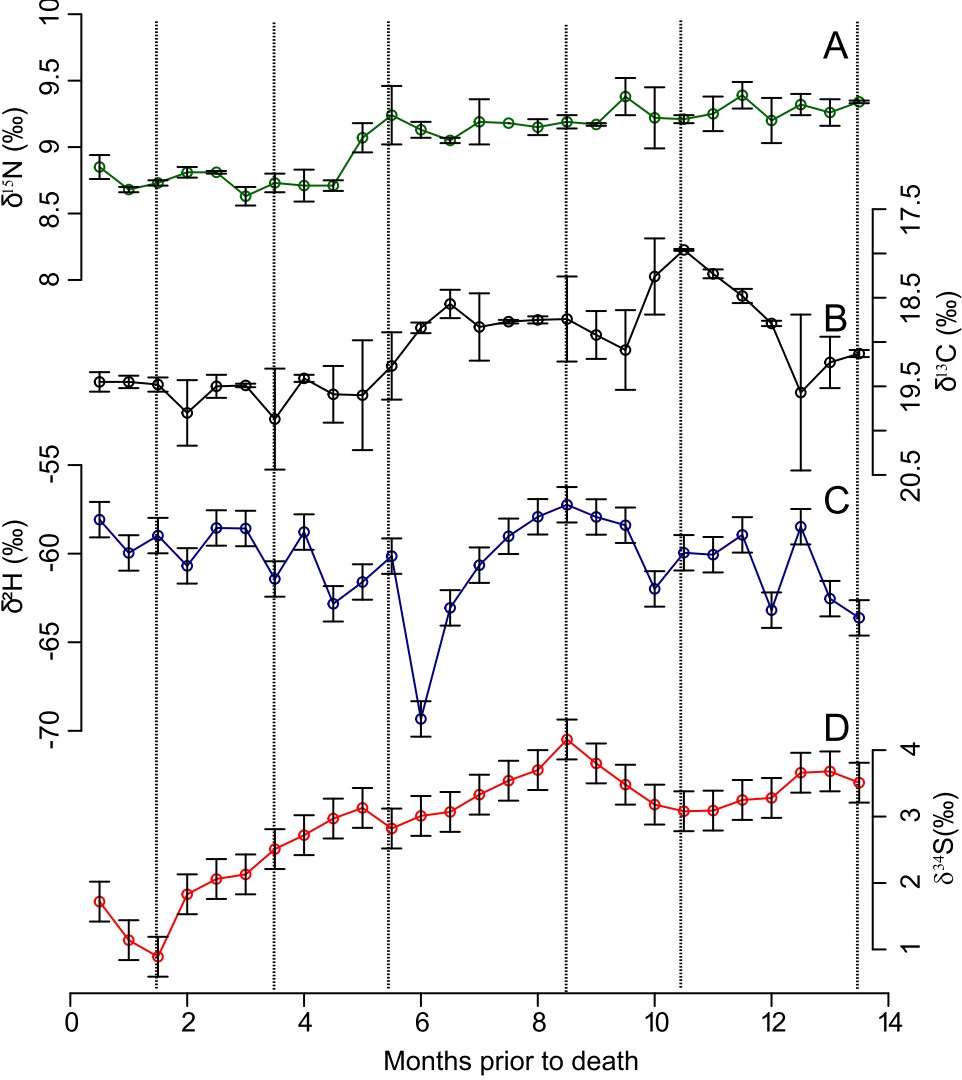

**Fig 5. Isotope values in Mr. Halifax's hair profile (complete data per hair segment available in the S2 Table). A.** $\delta^{15}N$ values; **B.** $\delta^{13}C$ values; **C.** $\delta^{2}H$ values; **D.** $\delta^{34}S$ values. Dashed lines define isotopic lines used for dual isotope probability maps in Fig 6. The R script to generate this figure is available as S2 Script. Data are available in S2 Table. Due to the difficulty in aligning the hair of Mr. Halifax, the timeframe represented by hair length is more elastic than usual [11, 36]. We used an approximate hair growth rate of 1cm per month for visualization.

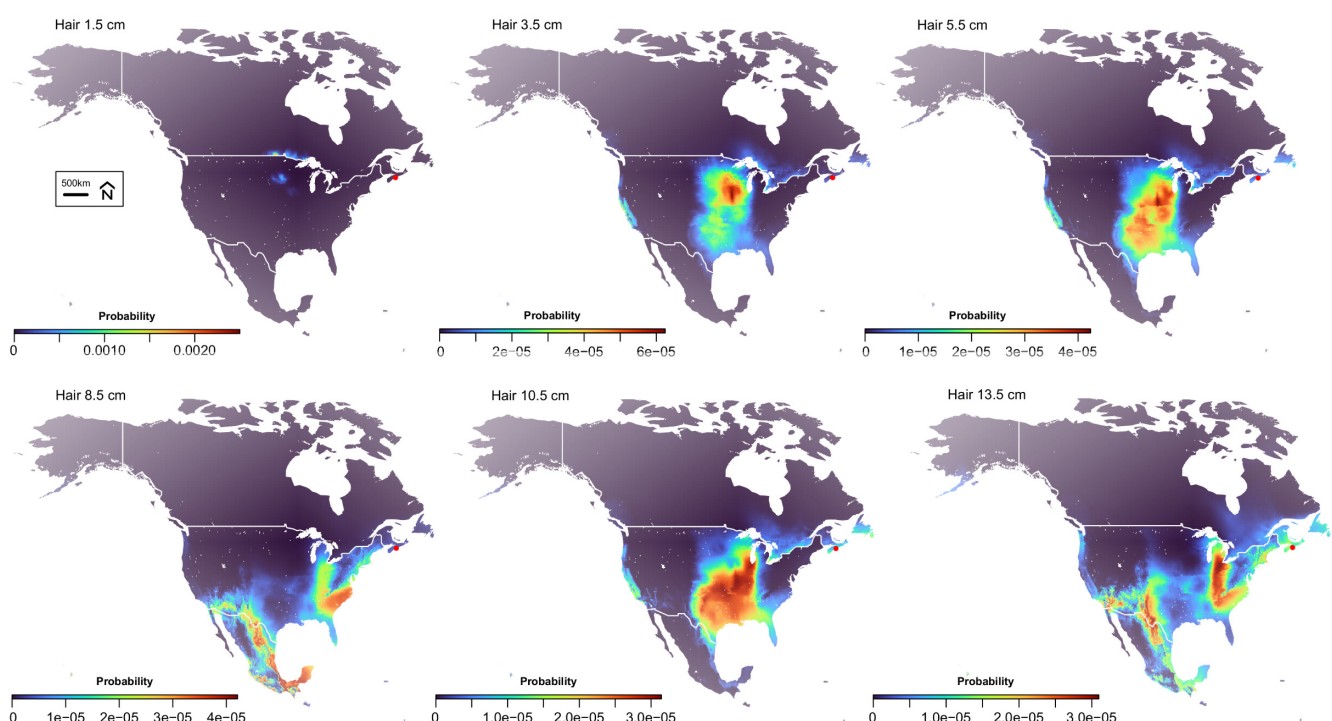

**Fig 6. Isotope-based probabilistic maps of provenance for Mr. Halifax estimated from $\delta^2H$ and $\delta^{34}S$ values for different representative hair segments.**
Locations where the human remains were discovered (Halifax, NS) are indicated by red points. Colours on the maps depict the predicted probability of origin. Coastlines and country boundaries are from http://www.naturalearthdata.com/. The R script to generate these figures is available in S2 Script. The data are available in S2 Table.

and the Prairies. However, the $\delta^{13}C$ values are higher between 5–12 months PTD and would be compatible with eastern Canada or the eastern US during this interval (Fig 5B). The $\delta^2H$ values of Mr. Halifax's hair ranged between −57.2 ‰ and −69.3 ‰. Compared to the $\delta^2H$ values of North American residents, Mr. Halifax's hair $\delta^2H$ value is most similar to the $\delta^2H$ values of hair from the eastern Canadian provinces, including Ontario, Quebec, the Maritimes, and Newfoundland (Table 2). Except for one very low value at 6 months PTD, the $\delta^2H$ values from Mr. Halifax's hair become more positive between 5–12 months PTD (Fig 5C).

We used these isotope values to generate single and dual-isotope probabilistic geographic maps from each segment of hair. We summarized in Fig 6 the map results by showing 6 individual segments corresponding to dotted lines in Fig 5. Dual $\delta^2H$ and $\delta^{34}S$ values from the base of the hair (1.5 cm, 3.5 cm and 5.5 cm) are compatible with the Canadian urban centers around the Great Lakes from Winnipeg to Sudbury and with US Midwest cities along the Mississippi river valley. At 8.5 months PTD, dual-isotope values are compatible with the southeastern US including Florida but are not compatible with regions in Canada. At 13.5 months PTD, the dual-isotope values are compatible with the Canadian Maritimes including Nova Scotia and Newfoundland but also with a broad region in the southeastern and southern US.

# Discussion

## Isocapes

Since the seminal work of Ehleringer et al. [33], country-scale studies analyzing isotopes in human hair of residents and/or tap water have become increasingly available providing the

basis to develop isoscapes in many regions of the world [33, 43–45, 60–63]. Efforts to harmonize isotope analyses have also facilitated the integration of data generated from different laboratories, countries and times [37].

**Sulfur isoscape.** As observed in previous studies [38, 42, 43, 45], the $\delta^{34}$S values in human hair are distinct for different countries across North America. Average $\delta^{34}$S values are 1.7 ± 1.0 ‰, 3.4 ± 1.1 ‰, and 4.4 ± 1.0 ‰ for Canada, the US, and Mexico, respectively. Different countries have different food production systems, supply chains and dietary habits that give rise to distinct baseline $\delta^{34}$S values [43]. Within countries, particularly Canada and the US, we observe a trend between $\delta^{34}$S hair values and distance to the coast. $\delta^{34}$S in North American crops should reflect the mixture of isotopically light sulfates from the soil solution, and isotopically heavy marine aerosols [38, 43, 45, 64, 65]. As food systems become more distant from the coast, bedrock sulfur or anthropogenic sources, which tend to have lower values, dominate decreasing $\delta^{34}$S values. The type of underlying geology can also influence the $\delta^{34}$S values of food systems locally [35]. For example, the presence of volcaniclastic sediments with low $\delta^{34}$S values in central Mexico or the low $\delta^{34}$S values in igneous and sedimentary rocks in interior Canada and USA likely contribute to the low $\delta^{34}$S values in local food systems [38, 43, 56]. The $\delta^{34}$S variability in hair from North American residents follows the isotopic pattern in food systems likely because customers obtain a large part of their sulfur from locally sourced high-protein food items (e.g., meat, yogurt, cheese, eggs). Even though North American residents have a supermarket diet and homogenized dietary habits, the regional patterns in $\delta^{34}$S in ecosystems incorporated into human hair highlights the importance of regional food supply in sulfur intake. Eating locally has become a more powerful movement in the last decade where locally produced foods have become more available and prized. The difference in sea salt aerosol deposition between North American countries might also explain their distinct average hair $\delta^{34}$S values and the trend in $\delta^{34}$S values with latitude. Mexico receives more marine sulfate than the US and Canada, as on average populations are closer to the coast in Mexico. The $\delta^{34}$S isoscape developed from this regression kriging approach shows excellent predictive power and strong spatial patterns that are promising for solving cross-border cold cases.

**Hydrogen isoscape.** As observed in previous studies [33, 60, 66], the $\delta^{2}$H values in human hair follow a continuous gradient across North America (Fig 2A). When $\delta^{2}$H hair data are corrected to be on the same reference scale [37], we can account for 80% of the variations in $\delta^{2}$H hair across North America using only the isotopic signal from modeled precipitation (Fig 2B). This strong relationship between local water and hair $\delta^{2}$H values reflects the local nature of water sources in human societies [33]. Drinking water typically has local to regional origin depending on the source [61–63]. Similarly, most beverages are derived from local to regional water sources including bottled water, coffee, tea or even soft drinks and milk [67]. Some of the residuals between precipitation and $\delta^{2}$H values in hair likely derive from differences between the $\delta^{2}$H of local precipitation and that of the consumed tap water. Tap water from different sources at one location might have different isotopic compositions. For example, tap water from shallow groundwater usually reflects an average of local precipitation $\delta^{2}$H [68] whereas tap water from deep aquifers, lakes or river waters might have more distant or fossil sources (e.g., glacial water) and/or might have been evaporated [61, 68]. This isotopic difference between water sources is exacerbated at the continental scale because different countries and regions rely on different water sources. For example, Mexico supplies the majority of its tap water from groundwater sources whereas Canada and the US primarily use surface water sources. Differences in dietary choices (i.e., food type), tap water sources and beverage consumption between participants living in the same area can also contribute to distinct hair $\delta^{2}$H values at a single location [67]. Many other factors can influence the relationship between $\delta^{2}$H in precipitation and in human hair, including the consumption of imported drinks with non-

local $\delta^2$H values, the presence of non-local study participants, or uncertainties in the predicted precipitation $\delta^2$H values. Despite those limitations, the continental $\delta^2$H isoscape shows excellent predictive power and strong spatial patterns that are promising for solving cross-border cold cases.

## Multi-isotope provenancing in international forensic studies

Stable isotopes have shown promise for provenancing unidentified decedents from cold-cases investigated by local jurisdictions [7, 9, 17, 21, 69–71] or to identify remains recovered from the sites of past wars and conflicts [8, 16, 40, 41, 69, 72]. However, the development of international isotope baselines from modern residents offers new possibilities to use isotopes for contemporary cross-border forensic applications [8, 10, 72]. In our globalized world, many forensic questions have international components. Through two case studies, we show that using dual $\delta^2$H and $\delta^{34}$S provenancing provides critical information to facilitate international forensic collaborations.

A lack of documents, fingerprints, dental and medical records, and family members to obtain family reference samples or antemortem samples for DNA comparison all present a challenge to identify UBCs at the US-Mexico border. These difficulties are further enhanced by the inability to narrow down the search area to more probable options. The four UBCs studied here were tentatively identified using other information, and serves as a basis to test the use of dual-isotope provenancing for identification purposes. Out of the 4 UBCs studied, none showed a local origin in the western US. All the individuals have dual-isotope values that place their origin in more southern regions of North America. As we only analysed isotope values in bulk hair for these individuals, we were not expecting precise provenancing because the isotope data in hair could reflect a mixture of isotope values from their last location of reference with isotope values inherited during their journey to the border. Specifically, the durations of travel can vary greatly, from merely days to up to months, and in rare cases even years. This largely depends on the availability of funds, previous extortions through cartels, previous (negative) experiences, place of origin, health status and, among many other variables personal decision-making. Even with this temporal resolution limitation, the dual-isotope data show promising results for these cases. Both UBC #1 and UBC #4 show high isotope-based probabilities in the city/region of inferred origin of the UBCs. UBC #2 likely resided for several months in the U.S. before being deported, re-entering and perishing in the US, explaining the strong signal from the southern US. The origin of UBC #3 could not be properly inferred because our predictions did not include Guatemala. Analyzing isotope data sequentially along the hair and from other tissues (e.g., teeth and bones) could thus substantially help reconstruct a more detailed travel history for those individuals. In contrast to hair which forms a few months prior to death, bones and teeth can preserve the isotope signature of locations where the individual resided earlier in the individuals' life. Those tissues could be a better recorder than hair to identify the place of residence or birth of these individuals.

While Mr. Halifax was found near an international airport, the forensic inquiry around this cold-case has remained national. The dual $\delta^2$H and $\delta^{34}$S measurements reveal that this individual traveled across large distances in the year prior to his death. Earlier in his life (i.e., 14 to 5 months PTD), the dual-isotope provenancing indicated travel and/or residence in the southeastern US. Dual isotope probability maps between 14 to 12 months PTD show some potential Canadian origin in coastal Nova Scotia or Newfoundland but are mostly compatible with the southeastern US. During the 12 to 5 months PTD interval, some probability maps of provenance are incompatible with any region in Canada. During the few months PTD, the individual lived either somewhere in eastern Canada, very likely in a municipality along the shore of

the Great Lakes such as Sudbury, Sault-Saint-Marie or Winnipeg, or in the northern US Midwest region. The carbon isotope data further support the proposed dual-isotope geographic assignments. While it is challenging to use carbon isotope for producing probabilistic maps of provenance because of fractionation associated with individual physiology and diet, the $\delta^{13}$C trend within a hair profile does record mobility [36]. In the case of Mr. Halifax, carbon isotope trends validate the dual $\delta^2$H and $\delta^{34}$S geographic assignments. Between 14 to 5 months PTD, higher $\delta^{13}$C values earlier in life culminating at −18 ‰ are more typical of the US. Low $\delta^{13}$C values (~ −19 ‰) a few months PTD are compatible with regions of Ontario around the Great Lakes (i.e., western Ontario and Manitoba). Even the $\delta^{15}$N values, which are usually not sensitive to mobility within a given country [36], appear to record a shift from higher to lower $\delta^{15}$N values around 5 months PTD. The higher $\delta^{15}$N values a year PTD possibly reflect a shift in dietary habits between places of residence (e.g., more fish consumption in the southeastern US leading to higher $\delta^{15}$N values).

This study demonstrates the potential of using multi-isotope analyses, and isotope-based geographic assignments as a rapid response tool in international cold-cases. The power of isotope provenancing increases several folds when multiple isotope systems are combined. The precision of dual-isotope results varies, but in some cases, the potential areas of origin that are identified are very specific. Reconstructing the movement history and origin with isotope data provide critical leads to law enforcement agencies for cross-border collaborations. The case of UBC #4 demonstrates the importance of large-scale and continuous isoscapes that are as spatially extensive as possible. This not only applies to deceased undocumented border crossers at the US-Mexico border but also to any other case of unidentified human remains with a potentially international outlook. Incomplete isoscapes bear the risk of exclusion or inclusion of regions that may otherwise be ex/included. This can severely affect the investigative work and thus slow down the identification process.

## Conclusion

In this work, we demonstrated that developing continental-scale isoscapes for $\delta^2$H and $\delta^{34}$S measurements in hair are feasible and yields accurate models with strong predictive potential for isotope-based provenancing. As more isotope data from human tissues are published and integrated, those models will become increasingly accurate and useful for provenance. These isoscapes are particularly useful to provide evidence for cross-border mobility in unidentified individuals. Through a first case study, we show that probabilistic maps of provenance based on the results of $\delta^2$H and $\delta^{34}$S analyses of bulk hair samples taken from deceased UBCs found at the US-Mexico border could help to identify their country/region of origin. Through a second case study, we show that sequential multi-isotope analysis along a hair strand combined with continuous probabilistic maps of provenance yields a detailed travel history of an unidentified individual between eastern Canada and the southeastern US. This type of evidence indicating international travel is essential to engage collaborations between law enforcement agencies. Now that the isotope baselines are established, we argue that multi-isotope profiling in human hair combined with isotope-based probabilistic provenancing provide a rapid, practical, inexpensive and powerful tool to screen current unknown deceased and cold-cases for potential international leads.

## Supporting information

**S1 Database. Sulfur isotope compilation from hair of North American residents.** Database containing the compiled and newly generated sulfur isotope data in hair of residents from North America. The database is formatted for compatibility with the accompanying R scripts

and assignR package *suborigin()* function [59]. See suborigin() documentation for column names.
(CSV)

**S2 Database. Hydrogen isotope compilation from hair of North American residents.** Database containing the compiled and newly generated hydrogen isotope data in hair of residents from North America. The database is formatted for compatibility with the accompanying R scripts and assignR package *suborigin()* function [59]. See suborigin() documentation for column names.
(CSV)

**S3 Database. Combined hydrogen and sulfur isotope compilation from hair of North American residents.** Database containing all the samples with combined hydrogen and sulfur isotope data in hair of residents from North America. The database is formatted for compatibility with the accompanying R scripts and assignR package *suborigin()* function [59]. See suborigin() documentation for column names.
(CSV)

**S1 Table. Sequential isotope data from Mr. Halifax.** Columns include the sample ID, hair length segment and corresponding isotope values and analytical precision for sulfur isotopes ("s34", "s34.SD"), hydrogen isotopes ("d2H", "d2H.SD"), carbon isotopes ("d13C", "d13C. SD") and nitrogen isotopes ("d15N", "d15N.SD") The table is formatted for compatibility with the accompanying R scripts.
(CSV)

**S2 Table. Individual isotope data from UBC individuals.** Columns include the sample ID and corresponding isotope values and analytical precision for sulfur isotopes ("s34", "s34.SD") and hydrogen isotopes ("d2H", "d2H.SD"). The table is formatted for compatibility with the accompanying R scripts.
(CSV)

**S1 Script. R script to generate sulfur isoscape in hair across North America.**
(R)

**S2 Script. R script to generate test the quality of sulfur and hydrogen isoscapes in hair across North America and to generate probabilistic maps.**
(R)

## Acknowledgments

The authors thank the anonymous hair donors for supporting this research. We also thank Paul Middlestead and the University of Ottawa's Jan Veizer Stable Isotope Laboratory for assistance in stable isotope analysis, and two anonymous reviewers and the editor for helpful comments.

## Author Contributions

**Conceptualization:** Clément P. Bataille, Saskia T. M. Ammer, Gilles St-Jean.

**Data curation:** Clément P. Bataille, Saskia T. M. Ammer.

**Formal analysis:** Clément P. Bataille, Saskia T. M. Ammer, Shelina Bhuiyan, Michelle M. G. Chartrand, Gilles St-Jean, Gabriel J. Bowen.

**Funding acquisition:** Clément P. Bataille, Saskia T. M. Ammer, Michelle M. G. Chartrand, Gilles St-Jean.

**Investigation:** Clément P. Bataille, Saskia T. M. Ammer.

**Methodology:** Clément P. Bataille, Saskia T. M. Ammer.

**Project administration:** Clément P. Bataille, Saskia T. M. Ammer, Michelle M. G. Chartrand.

**Resources:** Clément P. Bataille.

**Software:** Clément P. Bataille, Gabriel J. Bowen.

**Supervision:** Clément P. Bataille, Gilles St-Jean.

**Validation:** Clément P. Bataille.

**Visualization:** Clément P. Bataille.

**Writing – original draft:** Clément P. Bataille.

**Writing – review & editing:** Clément P. Bataille, Saskia T. M. Ammer, Shelina Bhuiyan, Michelle M. G. Chartrand, Gilles St-Jean, Gabriel J. Bowen.

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
