## [Decision Letter · Decision Letter 0]

7 Jun 2022

PONE-D-22-13353Multi-isotopes in Human Hair: A tool to initiate Cross-border Collaboration in International Cold-CasesPLOS ONE

Dear Dr. Bataille,

Thank you for submitting your manuscript to PLOS ONE. After careful consideration, we feel that it has merit but does not fully meet PLOS ONE’s publication criteria as it currently stands. Therefore, we invite you to submit a revised version of the manuscript that addresses the points raised during the review process.

Both reviewers underlined the quality and interest of the submitted manuscript. Reviewer 2 raises a relevant point on the use of international and/or in-house standards to verify the validity of inter-laboratory comparisons. This needs to be added to the material and method section. Information on the time range recorded by the hair in general (in comparison to other tissues) and the sampling pattern used for the study in particular is pertinent and should be also added. Additional minor corrections are listed in both reviews.

We look forward to receiving your revised manuscript.

Kind regards,

Dorothée Drucker

Academic Editor

PLOS ONE

Journal Requirements:

"This project was supported by the Chemical, Biological, Radiological-Nuclear and

Explosives Research and Technology Initiative [Award No. CRTI08-0116RD] awarded to

G.S.J., the Canadian Security and Safety Program Targeted Investment [CSSP-2018-TI2385] awarded to C.P.B. This research was further supported by the American Academy of

Forensic Sciences (AAFS) Humanitarian and Human Rights Resource Center (HHRRC)

grant, supported by the AAFS and National Institute of Justice (U.S. Department of Justice),

awarded to S.T.M.A. Work by G.J.B. was supported by US National Science Foundation

grants DBI-1565128 and DBI-1759730. The authors would like to thank the anonymous hair

donors for supporting this research. The authors would also like to thank Paul Middlestead

and the U of Ottawa's Jan Veizer Stable Isotope Laboratory for assistance in stable isotope

analysis. "

"This project was supported by the Chemical, Biological, Radiological-Nuclear and Explosives Research and Technology Initiative [Award No. CRTI08-0116RD] awarded to G.S.J., the Canadian Security and Safety Program Targeted Investment [CSSP-2018-TI-2385] awarded to C.P.B. This research was further supported by the American Academy of Forensic Sciences (AAFS) Humanitarian and Human Rights Resource Center (HHRRC) grant, supported by the AAFS and National Institute of Justice (U.S. Department of Justice), awarded to S.T.M.A. Work by G.J.B. was supported by US National Science Foundation grants DBI-1565128 and DBI-1759730. "

4. We note that Figures 1, 2, 4, and 6 in your submission contain [map/satellite] images which may be copyrighted. All PLOS content is published under the Creative Commons Attribution License (CC BY 4.0), which means that the manuscript, images, and Supporting Information files will be freely available online, and any third party is permitted to access, download, copy, distribute, and use these materials in any way, even commercially, with proper attribution. For these reasons, we cannot publish previously copyrighted maps or satellite images created using proprietary data, such as Google software (Google Maps, Street View, and Earth). For more information, see our copyright guidelines: http://journals.plos.org/plosone/s/licenses-and-copyright.

a. You may seek permission from the original copyright holder of Figures 1, 2, 4, and 6 to publish the content specifically under the CC BY 4.0 license.  

Reviewers' comments:

Reviewer's Responses to Questions

**Comments to the Author**

1. Is the manuscript technically sound, and do the data support the conclusions?

Reviewer #1: Yes

Reviewer #2: Yes

2. Has the statistical analysis been performed appropriately and rigorously? 

Reviewer #1: I Don't Know

Reviewer #2: Yes

3. Have the authors made all data underlying the findings in their manuscript fully available?

Reviewer #1: Yes

Reviewer #2: Yes

4. Is the manuscript presented in an intelligible fashion and written in standard English?

Reviewer #1: Yes

Reviewer #2: Yes

5. Review Comments to the Author

Reviewer #1: The present manuscript demonstrates the importance of international cooperation in elucidating the identity of unknown dead. The isoscapes produced illustrate the regional differences in the isotopic values of sulphur and hydrogen in hair samples from the North American population and are an important tool for getting closer to the region of origin of UBCs and other unknown individuals. In principle, however, when creating isoscapes it must be ensured that the analytical data of all database samples are comparable within the limits of measurement uncertainty. In the isotopic analysis of human hair, calibration of all database samples against the reference values of the currently available internationally recognised hair standards (USGS-42 and USGS-43) is essential.

It is not clear from the manuscript whether the isotopic values of all hair samples were calibrated against the USGS-42 and USGS-43 reference values before the isoscapes were created. Does this also apply to the sulphur isotope data of the Mexican hair samples taken from Ammer et al. 2020, because the original literature does not contain any precise information on this? Indeed, inter-laboratory comparisons show that the sulphur isotope data in hair in different laboratories can differ by several ‰ if no official or internal hair standards are taken into account as reference values.

For the regional classification of the individuals, isotope signature of their hair samples were used. The bulk analysis of hair from the UBCs yields mean values that contain information on the food and beverages consumed during the last 4 months of life. During this period of life, the individuals probably no longer stayed in their home region, but were "travelling" and exposed to more or less random food. These mean values are thus hardly compatible with the individuals' regions of origin, and may also feign stays in regions where they have never been. Ultimately, the informative value of the hair is limited to the time before death, and with the bulk analysis of a hair sample of 4 cm, it is not possible to differentiate the whereabouts (or changes of diet, health problems or starvation) during the last weeks of life, as for example with UBC#2.

Segmental analyses of the hair samples might have improved the regional allocations. A more precise regional assignment could have been expected from the examination of bones or teeth of the individuals.

Isotopic examination of these body tissues of UBCs should be the next step to get closer to their identity with the help of the isoscapes created in this work.

My special comments on the manuscript are in the PDF version.

Reviewer #2: Overall, this is an excellent paper and makes good use of existing reference data to test out on real unidentified remains cases, including a case study from Canada (where the decedent likely spent time in the US) and case studies of deceased undocumented border crossers. This paper is in excellent shape and only needs minor edits. It will make a great contribution to PLOS One and to forensic isotope literature.

Minor comments and edits:

Page 3, line 20: change "compose" to "are assimilated into keratin"

Page 3, line 25: change "it's hair inherits" to "their hair incorporates"

Page 3, line 31: change "inherited" to "assimilated"

Page 3: line 34: add values after delta-34S

Page 3, line 36: change "also composing" to "are also incorporating"

Page 4, line 4: change "in average" to "on average"

Page 4, line 5: soybeans are C3 plants (not C4)

Page 5, line 7: change to "US Border Patrol"

Page 5, line 19: change "isotope" to "isotopes"

Page 7, lines 5-6: change "Canada demonstrating" to "Canada would demonstrate"

Page 8, line 5: change "Americans" to "US Americans"

Page 10, line 20: change to "with larger datasets"

Page 10, line 21: change "approach" to "approaches"

Page 11, line 1: change to "dual isotope probabilistic maps"

In several places, "data is" is used and it should be "data are". Check to make sure plural phrasing is used. When discussing border crossers, please add the word deceased so it's clear that the samples are from decedents, not the living.

6. PLOS authors have the option to publish the peer review history of their article (what does this mean?). If published, this will include your full peer review and any attached files.

Reviewer #1: No

Reviewer #2: No

---

## [Author Response · Author response to Decision Letter 0]

3 Aug 2022

PONE-D-22-13353

Multi-isotopes in Human Hair: A tool to initiate Cross-border Collaboration in International Cold-Cases

PLOS ONE

Dear Dr. Bataille,

Thank you for submitting your manuscript to PLOS ONE. After careful consideration, we feel that it has merit but does not fully meet PLOS ONE’s publication criteria as it currently stands. Therefore, we invite you to submit a revised version of the manuscript that addresses the points raised during the review process.

Both reviewers underlined the quality and interest of the submitted manuscript. Reviewer 2 raises a relevant point on the use of international and/or in-house standards to verify the validity of inter-laboratory comparisons. This needs to be added to the material and method section. Information on the time range recorded by the hair in general (in comparison to other tissues) and the sampling pattern used for the study in particular is pertinent and should be also added. Additional minor corrections are listed in both reviews.

We look forward to receiving your revised manuscript. 

Kind regards,

Dorothée Drucker

Academic Editor

PLOS ONE

Journal Requirements:

We have ensured that our manuscript followed all the PLOS ONE’s style requirements including for file naming

We have provided this information by adding the “informed written consent”. Our study did not include minors.

"This project was supported by the Chemical, Biological, Radiological-Nuclear and

Explosives Research and Technology Initiative [Award No. CRTI08-0116RD] awarded to

G.S.J., the Canadian Security and Safety Program Targeted Investment [CSSP-2018-TI2385] awarded to C.P.B. This research was further supported by the American Academy of

Forensic Sciences (AAFS) Humanitarian and Human Rights Resource Center (HHRRC)

grant, supported by the AAFS and National Institute of Justice (U.S. Department of Justice),

awarded to S.T.M.A. Work by G.J.B. was supported by US National Science Foundation

grants DBI-1565128 and DBI-1759730. The authors would like to thank the anonymous hair

donors for supporting this research. The authors would also like to thank Paul Middlestead

and the U of Ottawa's Jan Veizer Stable Isotope Laboratory for assistance in stable isotope

analysis. "

"This project was supported by the Chemical, Biological, Radiological-Nuclear and Explosives Research and Technology Initiative [Award No. CRTI08-0116RD] awarded to G.S.J., the Canadian Security and Safety Program Targeted Investment [CSSP-2018-TI-2385] awarded to C.P.B. This research was further supported by the American Academy of Forensic Sciences (AAFS) Humanitarian and Human Rights Resource Center (HHRRC) grant, supported by the AAFS and National Institute of Justice (U.S. Department of Justice), awarded to S.T.M.A. Work by G.J.B. was supported by US National Science Foundation grants DBI-1565128 and DBI-1759730. "

We have removed the funding information from the acknowledgments. The funding information above are correct and contain all the information. We have added it to the cover letter.

4. We note that Figures 1, 2, 4, and 6 in your submission contain [map/satellite] images which may be copyrighted. All PLOS content is published under the Creative Commons Attribution License (CC BY 4.0), which means that the manuscript, images, and Supporting Information files will be freely available online, and any third party is permitted to access, download, copy, distribute, and use these materials in any way, even commercially, with proper attribution. For these reasons, we cannot publish previously copyrighted maps or satellite images created using proprietary data, such as Google software (Google Maps, Street View, and Earth). For more information, see our copyright guidelines: http://journals.plos.org/plosone/s/licenses-and-copyright.

This is incorrect. None of the figures contain any proprietary information. All the figures were designed under CC License. The images are not satellite images they are probability maps generated by the R script given in supplementary material. All the images can be published under the CC BY 4-0. The country polygons on the maps are from Natural Earth. We added a mention: “Coastlines and country boundaries are from http://www.naturalearthdata.com/” in Figs 1,2, 4 and 6.

a. You may seek permission from the original copyright holder of Figures 1, 2, 4, and 6 to publish the content specifically under the CC BY 4.0 license. 

We have modified the file naming and added the requested captions to the supporting information.

We have verified the reference list. 

Reviewers' comments:

Reviewer's Responses to Questions

Comments to the Author

1. Is the manuscript technically sound, and do the data support the conclusions?

Reviewer #1: Yes

Reviewer #2: Yes

2. Has the statistical analysis been performed appropriately and rigorously? 

Reviewer #1: I Don't Know

Reviewer #2: Yes

3. Have the authors made all data underlying the findings in their manuscript fully available?

Reviewer #1: Yes

Reviewer #2: Yes

4. Is the manuscript presented in an intelligible fashion and written in standard English?

Reviewer #1: Yes

Reviewer #2: Yes

5. Review Comments to the Author

Reviewer #1: The present manuscript demonstrates the importance of international cooperation in elucidating the identity of unknown dead. The isoscapes produced illustrate the regional differences in the isotopic values of sulphur and hydrogen in hair samples from the North American population and are an important tool for getting closer to the region of origin of UBCs and other unknown individuals. In principle, however, when creating isoscapes it must be ensured that the analytical data of all database samples are comparable within the limits of measurement uncertainty. In the isotopic analysis of human hair, calibration of all database samples against the reference values of the currently available internationally recognised hair standards (USGS-42 and USGS-43) is essential.

It is not clear from the manuscript whether the isotopic values of all hair samples were calibrated against the USGS-42 and USGS-43 reference values before the isoscapes were created. Does this also apply to the sulphur isotope data of the Mexican hair samples taken from Ammer et al. 2020, because the original literature does not contain any precise information on this? Indeed, inter-laboratory comparisons show that the sulphur isotope data in hair in different laboratories can differ by several ‰ if no official or internal hair standards are taken into account as reference values.

We thank the author for this excellent and critical comment. We have rewritten most of the method isotope section to detail the traceability of δ2H and δ34S isotope data. We added some information to demonstrate that all samples are traceable to the same scale. For δ34S, all results are traceable to the Vienna Canyon Diablo Triolite (VCDT) scale via IAEA-S-1, IAEA-S-2 and IAEA-S-3. We wanted to point to the reviewer that USGS42 and USGS43 are themselves calibrated to IAEA-S-1, IAEA-S-2 and IAEA-S-3. Data from Mexico were analyzed for δ34S values at UC Davis Stable Isotope Facility, which used 6 internal RMs including keratin calibrated against IAEA-S-1, IAEA-S-2 and IAEA-S-3. Data from the USA was obtained from the SIRFER Laboratory at the University of Utah and were calibrated to 3 internal RMs calibrated using IAEA-S-1, IAEA-S-2 and IAEA-S-3. As a quality check, we asked SIRFER to rerun new USGS42 and USGS43 using the same analytical protocol and the obtained values for USGS42 (7.94±0.06‰ (n=5)) and USGS43 (10.37±0.13‰ (n=5)) compare well with the expected values for USGS42 (7.84 ± 0.25 ‰) and USGS43 (10.46 ± 0.22 ‰). Again, as USGS42 and USGS43 are themselves calibrated against IAEA-S-1, IAEA-S-2 and IAEA-S-3, this analysis technically superfluous. Data from Canada (and the forensic cases) was obtained from the Veizer Stable Isotope Lab at the University of Ottawa. RMs used for calibration were IAEA-S-1 (−0.3 ‰), IAEA-S-2 (22.7 ‰) and IAEA-S-3 (−32.6 ‰). The values used for IAEA-S-2 and IAEA-S-3 were not the same as used in the other laboratories. To verify these measurement are compatible with the other measurements, both USGS42 and USGS43 were analyzed against IAEA-S-1, IAEA-S-2 and IAEA-S-3, and the measured δ34S values and one standard deviation (7.58 ± 0.13 ‰ (n=3; USGS42) and 10.22 ± 0.15 ‰ (n=3; USGS43)) overlapped with the certified δ34S values and uncertainties for USGS42 and USGS43.RMs used for calibration were directly IAEA-S-1, IAEA-S-2 and IAEA-S-3 making them directly comparable to other dataset. Again to validate the approach, we analyzed USGS42 and USGS43 as QC and obtained measured values that overlapped with the certified δ34S values and uncertainties for these standards. 

For δ2H values, data from Mexico were analyzed at the Jan Veizer Stable Isotope Laboratory at the University of Ottawa using the comparative analysis approach. However, instead of using only CBS and KHS for calibration, a three-point calibration with CBS, KHS and USGS43 was used for generating these data. This three-point calibration has technically a slightly different equation as the two-point calibration using CBS and KHS only. However, the effect on the obtained δ2H values is minimal (<0.7‰) and negligible relative to analytical uncertainty. Additionally, USGS42 and USGS43 values were re-calibrated using the CBS and KHS comparative approach (Soto et al. 2017) and their values fall within analytical uncertainty of the certified values (Coplen et al. 2016). Consequently, USGS42 and USGS43 are also traceable to the VSMOW-SLAP scale. Finally, USGS42 was included as a quality check in all hair analysis sequences and the measured values were within the certified values and uncertainties. The δ2H data from the USA and Canada were analyzed using older protocols and calibration standards, and were recalibrated in a previous study (Magozzi et al. 2021). This study provides a high level of details on traceability. After uncertainty propagation the rescaled δ2H values have relatively large uncertainty ~3‰. Based on these facts, the δ2H data generated and compiled in this study are all traceable to the Vienna Standard Mean Ocean Water-Standard Light Antarctic Precipitation (VSMOW-SLAP) scale. 

For the regional classification of the individuals, isotope signature of their hair samples were used. The bulk analysis of hair from the UBCs yields mean values that contain information on the food and beverages consumed during the last 4 months of life. During this period of life, the individuals probably no longer stayed in their home region, but were "travelling" and exposed to more or less random food. These mean values are thus hardly compatible with the individuals' regions of origin, and may also feign stays in regions where they have never been. Ultimately, the informative value of the hair is limited to the time before death, and with the bulk analysis of a hair sample of 4 cm, it is not possible to differentiate the whereabouts (or changes of diet, health problems or starvation) during the last weeks of life, as for example with UBC#2.

Segmental analyses of the hair samples might have improved the regional allocations. A more precise regional assignment could have been expected from the examination of bones or teeth of the individuals.

We agree with this comment and we have added some sentences in the methods and in the discussion to underline that the bulk hair analyses is not the ideal approach. 

Isotopic examination of these body tissues of UBCs should be the next step to get closer to their identity with the help of the isoscapes created in this work.

Yes this has actually been done isotopes were analyzed in bones and teeth as well (see Saskia Ammer’s doctoral thesis) but we limited this paper to hair as we did not develop the bones and teeth isoscapes. We added a sentence to underline this point.

My special comments on the manuscript are in the PDF version.

We followed most of the recommendations of the reviewer in the pdf version including adding references, changing figure 4 and adding some small clarifications throughout the manuscript. The only comment we did not follow is to modify the hair growth rate to 1.4 cm per month. While this is true that 1.4 cm per month might be a better approximation of the growth rate, the age model of this individual is pretty elastic considering that we used a dreadlock and had difficulty aligning hair (this was mentioned in the method section). We added this detail in the figure caption: “Due to the difficulty in aligning the hair of Mr. Halifax, the timeframe represented by hair length is more elastic than usual [10,35]. We used an approximate hair growth rate of 1cm per month for visualization.” 

Reviewer #2: Overall, this is an excellent paper and makes good use of existing reference data to test out on real unidentified remains cases, including a case study from Canada (where the decedent likely spent time in the US) and case studies of deceased undocumented border crossers. This paper is in excellent shape and only needs minor edits. It will make a great contribution to PLOS One and to forensic isotope literature.

Minor comments and edits:

Page 3, line 20: change "compose" to "are assimilated into keratin"

We modified as suggested.

Page 3, line 25: change "it's hair inherits" to "their hair incorporates"

We modified as suggested.

Page 3, line 31: change "inherited" to "assimilated"

We modified as suggested.

Page 3: line 34: add values after delta-34S

We added as suggested.

Page 3, line 36: change "also composing" to "are also incorporating"

We modified as suggested.

Page 4, line 4: change "in average" to "on average"

We modified as suggested.

Page 4, line 5: soybeans are C3 plants (not C4)

We removed soybeans as suggested.

Page 5, line 7: change to "US Border Patrol"

We changed as suggested.

Page 5, line 19: change "isotope" to "isotopes"

We modified as suggested.

Page 7, lines 5-6: change "Canada demonstrating" to "Canada would demonstrate"

We modified as suggested.

Page 8, line 5: change "Americans" to "US Americans"

We modified as suggested.

Page 10, line 20: change to "with larger datasets"

We modified as suggested.

Page 10, line 21: change "approach" to "approaches"

We modified as suggested.

Page 11, line 1: change to "dual isotope probabilistic maps"

We modified as suggested.

In several places, "data is" is used and it should be "data are". Check to make sure plural phrasing is used. 

We modified as suggested.

When discussing border crossers, please add the word deceased so it's clear that the samples are from decedents, not the living.

We added as suggested.

6. PLOS authors have the option to publish the peer review history of their article (what does this mean?). If published, this will include your full peer review and any attached files.

Do you want your identity to be public for this peer review? For information about this choice, including consent withdrawal, please see our Privacy Policy.

Reviewer #1: No

Reviewer #2: No

---

## [Decision Letter · Decision Letter 1]

6 Sep 2022

PONE-D-22-13353R1Multi-isotopes in Human Hair: A tool to initiate Cross-border Collaboration in International Cold-CasesPLOS ONE

Dear Dr. Bataille,

Thank you for submitting your manuscript to PLOS ONE. After careful consideration, we feel that it has merit but does not fully meet PLOS ONE’s publication criteria as it currently stands. Therefore, we invite you to submit a revised version of the manuscript that addresses the points raised during the review process.

Dear Author(s), I had the chance to have a read through the paper also comparing the first and the revised version  and I really con see a sensitive development of the MS. Some minor comments arises from the reviewer and I also add some comments regarding the methodological part. I will be grateful if you can try to accomodate all the issues raised.

We look forward to receiving your revised manuscript.

Kind regards,

Fabio Marzaioli, Ph.D

Academic Editor

PLOS ONE

Journal Requirements:

Additional Editor Comments:

Dear Author(s)

I had the chance to have a read through the paper also comparing the first and the revised version (I entered later in the manuscript handling) and I really con see a sensitive development of the MS.

Anyway Ref. 1 was asking for some more work to really improve the paper.

In details Reviewer commented:

Abstract

Line37: please change „δ2H and δ34S measurements“ to „ δ2H and δ34S values“

Introduction

Line 80: please delete “to bi-monthly”. Chronological information of hair depend on the length of a hair segments, analyses of <0.5 mm segments with a 2-weekly resolution (and less) is possible (as you did for the hair samples from Mr. Halifax).

Line 81: The mean growth rate of human hair mostly is defined by 1 cm/ month. Please delete the growth rate of 0.7 mm/ month. To my knowledge, in literature this low growth rate was only mentioned in one single paper for hairs of negroid individuals, but this is most likely not the mean value for most of the human hair.

Line 82: please add reference 8 and 10.

Line 88: it is well known that C, N, H, O and S (not only H, O and S) are assimilated into hair keratin, because these are the components of the protein. References 33 and 34 should be mentioned together.

Line 97: please check the listed references. In several references δ2H in hair has not been analysed for geolocation. You may add δ18O and Sr to the sentence, or you may only list those in which δ2H has actually been analysed.

Line 117 f: This sentence could be deleted, if line 88 was changed as suggested.

Line 119: add reference 12.

Line 164: I’d like to thank the authors for their elaboration on the problem of calibrating the sulphur isotope values in the hair. However, I would like to note here again that δ34S analyses on hair samples (and especially on collagen samples) are very tricky and challenging. Therefore, a comparability check of the δ34S hair results between the UCD lab and the Veizer lab was necessary before establishing the δ34S isoscape.

In principle, especially for δ34S and δ2H analyses on human hair it is necessary to calibrate the measured values against internal reference standards that are composed of the same material as the measured sample. These reference standards could be made from human hair, or from horse or cattle tail hair, and should cover the entire range of the measured values. Thus, for a two-point, better a three point calibration, several hair standards from different parts of the world are needed which should run with each series of measurements. Furthermore, they must be recalibrated against international standards such as USGS42 and USGS43, which unfortunately do not cover the whole range of values especially with regard to sulphur (and hydrogen) isotope results.

Line 166: RMs = reference materials

Line 253: I assume that δ13C, δ15N and δ34S values of the UBCs hair were analysed at the UCD, because the values agree to those in Ammer’s thesis.

Figure 1 A and B. Please explain the abbreviation “hair-obs”.

Line 345: Pease add the references for the published δ34S data (also in Fig. 2).

Line 472: the lowest δ2H value was 6 months PTD

Line 506 f: please add that also specific geological conditions give rise to distinct baseline d34S values, in particular for Mexico the occurrence of volcanoes, leading to low δ34S values (see thesis of Ammer 2020). Even if this is not clearly visible on the maps due to the low resolution, this should be mentioned in that section.

Line 545 ff.: Not only the δ2H values of drinks, but also the δ2H values of the food most probably affect the hair δ2H values (according to the equation in Fig. 2 A). From that it can be concluded that most of the hydrogen in keratin is not from the drinking water.

Line 553: change ref 20 to ref 15.

Line 584 f: add: ... where the individual grew up and lived..., and better “the places of residence earlier in the individuals’ life” (because the place of birth cannot be identified)

Line 588 ff: For description of the living phases of Mr. Halifax I’d suggest to mention first the former and then the more recent months of life, e.g. during the 15 to 5 month PTD interval. The shift of the isotope values along the hair strand as a result of dietary changes is always from the older to the younger part of the hair. Consequently, the shift to higher δ13C values of ~-18 ‰ was between 12 and 10 months PTD, followed in turn by a decrease of the δ13C values between 10 and 5 months PTD (from that you may conclude that he possibly entered eastern US or eastern Canada about 12 months PTD). Low δ13C values (~-19 ‰) during the last 5 months PTD are compatible...

Line 607: For δ15N values in the hair strand I’d suggest to write: .... appear to record a shift from higher to lower δ15N values around 5 months PTD.

Line 631 ff: ...based on the results of δ2H and δ34S analyses of bulk hair samples taken from deceased UBCs found at the US-Mexico border could help to identify...

Line 639: Screening is also possible for current unknown deceased, not only for cold-cases.

Please note that based on the Iupac guidelines and according to Coplen 2011, “δ” should be written in italic font throughout the manuscript.

• Coplen, T. B. (2011). Guidelines and recommended terms for expression of stable‐isotope‐ratio and gas‐ratio measurement results. Rapid communications in mass spectrometry, 25(17), 2538-2560

Moreover from my side:

1) I would like to ask you to insert before utilizing delta notation its definition according to the last reference suggested by the reviewer (i.e. Coplen, T. B. (2011). Guidelines and recommended terms for expression of stable‐isotope‐ratio and gas‐ratio measurement results. Rapid communications in mass spectrometry, 25(17), 2538-2560).

2) I am curious regarding some data showed in the "Isotopic analysis of residents' hair samples from across North America" paragraph. In details for the second batch of d34S analysed in Venezuela, the authors did a sort of accuracy test (it cannot be called Quality Check because this term has a specific meaning according to sample QA/QC procedures) by comparing data measured on USGS42 and 43 with their certificates. I am a bit worried about reported uncertainties in the framework of this exercise. In details lab internal standards utilized and calibrated against IAEA S series (s1-s2-s2). Reported uncertainties affecting lab internal standards span from .3 to .4 per mil. My question regards the uncertainties affecting produced measured USGS values which are sensitively lower than the ones affecting the lab internal standards. It is a rule of thumb that measurements and data manipulation can only enlarge uncertainties but this is not the case. Can you please reply?

3) It is not clear to me where samples coming from [37] study have been analysed and why only tests performed on some analytical batches are presented (i.e. to what lab is the accuracy test referred?).

Thanks

Reviewers' comments:

Reviewer's Responses to Questions

**Comments to the Author**

1. If the authors have adequately addressed your comments raised in a previous round of review and you feel that this manuscript is now acceptable for publication, you may indicate that here to bypass the “Comments to the Author” section, enter your conflict of interest statement in the “Confidential to Editor” section, and submit your "Accept" recommendation.

Reviewer #1: All comments have been addressed

Reviewer #2: All comments have been addressed

2. Is the manuscript technically sound, and do the data support the conclusions?

Reviewer #1: Yes

Reviewer #2: Yes

3. Has the statistical analysis been performed appropriately and rigorously? 

Reviewer #1: I Don't Know

Reviewer #2: Yes

4. Have the authors made all data underlying the findings in their manuscript fully available?

Reviewer #1: Yes

Reviewer #2: Yes

5. Is the manuscript presented in an intelligible fashion and written in standard English?

Reviewer #1: Yes

Reviewer #2: Yes

6. Review Comments to the Author

Reviewer #1: The authors have done an excellent job of revising the manuscript and have thus once again significantly improved it. Below I have a few comments and suggestions that should be considered before publication:

Abstract

Line37: please change „δ2H and δ34S measurements“ to „ δ2H and δ34S values“

Introduction

Line 80: please delete “to bi-monthly”. Chronological information of hair depend on the length of a hair segments, analyses of <0.5 mm segments with a 2-weekly resolution (and less) is possible (as you did for the hair samples from Mr. Halifax).

Line 81: The mean growth rate of human hair mostly is defined by 1 cm/ month. Please delete the growth rate of 0.7 mm/ month. To my knowledge, in literature this low growth rate was only mentioned in one single paper for hairs of negroid individuals, but this is most likely not the mean value for most of the human hair.

Line 82: please add reference 8 and 10.

Line 88: it is well known that C, N, H, O and S (not only H, O and S) are assimilated into hair keratin, because these are the components of the protein. References 33 and 34 should be mentioned together.

Line 97: please check the listed references. In several references δ2H in hair has not been analysed for geolocation. You may add δ18O and Sr to the sentence, or you may only list those in which δ2H has actually been analysed.

Line 117 f: This sentence could be deleted, if line 88 was changed as suggested.

Line 119: add reference 12.

Line 164: I’d like to thank the authors for their elaboration on the problem of calibrating the sulphur isotope values in the hair. However, I would like to note here again that δ34S analyses on hair samples (and especially on collagen samples) are very tricky and challenging. Therefore, a comparability check of the δ34S hair results between the UCD lab and the Veizer lab was necessary before establishing the δ34S isoscape.

In principle, especially for δ34S and δ2H analyses on human hair it is necessary to calibrate the measured values against internal reference standards that are composed of the same material as the measured sample. These reference standards could be made from human hair, or from horse or cattle tail hair, and should cover the entire range of the measured values. Thus, for a two-point, better a three point calibration, several hair standards from different parts of the world are needed which should run with each series of measurements. Furthermore, they must be recalibrated against international standards such as USGS42 and USGS43, which unfortunately do not cover the whole range of values especially with regard to sulphur (and hydrogen) isotope results.

Line 166: RMs = reference materials

Line 253: I assume that δ13C, δ15N and δ34S values of the UBCs hair were analysed at the UCD, because the values agree to those in Ammer’s thesis.

Figure 1 A and B. Please explain the abbreviation “hair-obs”.

Line 345: Pease add the references for the published δ34S data (also in Fig. 2).

Line 472: the lowest δ2H value was 6 months PTD

Line 506 f: please add that also specific geological conditions give rise to distinct baseline d34S values, in particular for Mexico the occurrence of volcanoes, leading to low δ34S values (see thesis of Ammer 2020). Even if this is not clearly visible on the maps due to the low resolution, this should be mentioned in that section.

Line 545 ff.: Not only the δ2H values of drinks, but also the δ2H values of the food most probably affect the hair δ2H values (according to the equation in Fig. 2 A). From that it can be concluded that most of the hydrogen in keratin is not from the drinking water.

Line 553: change ref 20 to ref 15.

Line 584 f: add: ... where the individual grew up and lived..., and better “the places of residence earlier in the individuals’ life” (because the place of birth cannot be identified)

Line 588 ff: For description of the living phases of Mr. Halifax I’d suggest to mention first the former and then the more recent months of life, e.g. during the 15 to 5 month PTD interval. The shift of the isotope values along the hair strand as a result of dietary changes is always from the older to the younger part of the hair. Consequently, the shift to higher δ13C values of ~-18 ‰ was between 12 and 10 months PTD, followed in turn by a decrease of the δ13C values between 10 and 5 months PTD (from that you may conclude that he possibly entered eastern US or eastern Canada about 12 months PTD). Low δ13C values (~-19 ‰) during the last 5 months PTD are compatible...

Line 607: For δ15N values in the hair strand I’d suggest to write: .... appear to record a shift from higher to lower δ15N values around 5 months PTD.

Line 631 ff: ...based on the results of δ2H and δ34S analyses of bulk hair samples taken from deceased UBCs found at the US-Mexico border could help to identify...

Line 639: Screening is also possible for current unknown deceased, not only for cold-cases.

Please note that based on the Iupac guidelines and according to Coplen 2011, “δ” should be written in italic font throughout the manuscript.

• Coplen, T. B. (2011). Guidelines and recommended terms for expression of stable‐isotope‐ratio and gas‐ratio measurement results. Rapid communications in mass spectrometry, 25(17), 2538-2560

Reviewer #2: This manuscript is much improved over the first version. I believe the authors have addressed all the reviewer comments. I only had very minor technical edits in the attached PDF.

7. PLOS authors have the option to publish the peer review history of their article (what does this mean?). If published, this will include your full peer review and any attached files.

Reviewer #1: No

Reviewer #2: No

---

## [Author Response · Author response to Decision Letter 1]

14 Sep 2022

PONE-D-22-13353R1

Multi-isotopes in Human Hair: A tool to initiate Cross-border Collaboration in International Cold-Cases

PLOS ONE

Dear Dr. Bataille,

Thank you for submitting your manuscript to PLOS ONE. After careful consideration, we feel that it has merit but does not fully meet PLOS ONE’s publication criteria as it currently stands. Therefore, we invite you to submit a revised version of the manuscript that addresses the points raised during the review process.

Dear Author(s), I had the chance to have a read through the paper also comparing the first and the revised version and I really con see a sensitive development of the MS. Some minor comments arises from the reviewer and I also add some comments regarding the methodological part. I will be grateful if you can try to accomodate all the issues raised.

We look forward to receiving your revised manuscript. 

Kind regards,

Fabio Marzaioli, Ph.D

Academic Editor

PLOS ONE

Journal Requirements:

Additional Editor Comments:

Dear Author(s)

I had the chance to have a read through the paper also comparing the first and the revised version (I entered later in the manuscript handling) and I really con see a sensitive development of the MS.

Anyway Ref. 1 was asking for some more work to really improve the paper.

Moreover from my side:

1) I would like to ask you to insert before utilizing delta notation its definition according to the last reference suggested by the reviewer (i.e. Coplen, T. B. (2011). Guidelines and recommended terms for expression of stable‐isotope‐ratio and gas‐ratio measurement results. Rapid communications in mass spectrometry, 25(17), 2538-2560).

We have added an explanation of the delta notation definition in the introduction and referenced to the requested paper.

2) I am curious regarding some data showed in the "Isotopic analysis of residents' hair samples from across North America" paragraph. In details for the second batch of d34S analysed in Venezuela, the authors did a sort of accuracy test (it cannot be called Quality Check because this term has a specific meaning according to sample QA/QC procedures) by comparing data measured on USGS42 and 43 with their certificates. I am a bit worried about reported uncertainties in the framework of this exercise. In details lab internal standards utilized and calibrated against IAEA S series (s1-s2-s2). Reported uncertainties affecting lab internal standards span from .3 to .4 per mil. My question regards the uncertainties affecting produced measured USGS values which are sensitively lower than the ones affecting the lab internal standards. It is a rule of thumb that measurements and data manipulation can only enlarge uncertainties but this is not the case. Can you please reply?

We believe our writing might have been unclear and cause confusion on this point. The reported uncertainties are not a full error propagation of uncertainty in the calibrated values, they are simply one standard deviation of the replicate d34S measurements obtained from the analyses of USGS42 and USGS43 samples performed at the University of Utah. We compared the mean value to that of the certified values for these standards and showed that the measured mean values fall within the 1 sigma range of the certified value. To clarify, we have rewritten some sentences in this section to better characterize what each uncertainty represents. 

3) It is not clear to me where samples coming from [37] study have been analysed and why only tests performed on some analytical batches are presented (i.e. to what lab is the accuracy test referred?).

We have added some details in the method section about where and when the different analyses were performed. We added details about the extra analysis of USGS42 and USGS43 samples performed at the University of Utah. These extra analyses of USGS42 and 43 were performed to answer the comments of reviewer 1 during the first round of review. 

Thanks

Reviewers' comments:

Reviewer's Responses to Questions

Comments to the Author

1. If the authors have adequately addressed your comments raised in a previous round of review and you feel that this manuscript is now acceptable for publication, you may indicate that here to bypass the “Comments to the Author” section, enter your conflict of interest statement in the “Confidential to Editor” section, and submit your "Accept" recommendation.

Reviewer #1: All comments have been addressed

Reviewer #2: All comments have been addressed

2. Is the manuscript technically sound, and do the data support the conclusions?

Reviewer #1: Yes

Reviewer #2: Yes

3. Has the statistical analysis been performed appropriately and rigorously? 

Reviewer #1: I Don't Know

Reviewer #2: Yes

4. Have the authors made all data underlying the findings in their manuscript fully available?

Reviewer #1: Yes

Reviewer #2: Yes

5. Is the manuscript presented in an intelligible fashion and written in standard English?

Reviewer #1: Yes

Reviewer #2: Yes

6. Review Comments to the Author

Reviewer #1: The authors have done an excellent job of revising the manuscript and have thus once again significantly improved it. Below I have a few comments and suggestions that should be considered before publication:

Abstract

Line37: please change „δ2H and δ34S measurements“ to „ δ2H and δ34S values“

We corrected as suggested

Introduction

Line 80: please delete “to bi-monthly”. Chronological information of hair depend on the length of a hair segments, analyses of <0.5 mm segments with a 2-weekly resolution (and less) is possible (as you did for the hair samples from Mr. Halifax).

We removed as suggested

Line 81: The mean growth rate of human hair mostly is defined by 1 cm/ month. Please delete the growth rate of 0.7 mm/ month. To my knowledge, in literature this low growth rate was only mentioned in one single paper for hairs of negroid individuals, but this is most likely not the mean value for most of the human hair.

We corrected as suggested

Line 82: please add reference 8 and 10.

We corrected as suggested

Line 88: it is well known that C, N, H, O and S (not only H, O and S) are assimilated into hair keratin, because these are the components of the protein. References 33 and 34 should be mentioned together.

We modified as suggested.

Line 97: please check the listed references. In several references δ2H in hair has not been analysed for geolocation. You may add δ18O and Sr to the sentence, or you may only list those in which δ2H has actually been analysed.

We decided to change this list of citations to only one citation (e.g., Ehleringer et al. 2008)

Line 117 f: This sentence could be deleted, if line 88 was changed as suggested.

We removed that sentence

Line 119: add reference 12.

We added reference 12.

Line 164: I’d like to thank the authors for their elaboration on the problem of calibrating the sulphur isotope values in the hair. However, I would like to note here again that δ34S analyses on hair samples (and especially on collagen samples) are very tricky and challenging. Therefore, a comparability check of the δ34S hair results between the UCD lab and the Veizer lab was necessary before establishing the δ34S isoscape.

In principle, especially for δ34S and δ2H analyses on human hair it is necessary to calibrate the measured values against internal reference standards that are composed of the same material as the measured sample. These reference standards could be made from human hair, or from horse or cattle tail hair, and should cover the entire range of the measured values. Thus, for a two-point, better a three point calibration, several hair standards from different parts of the world are needed which should run with each series of measurements. Furthermore, they must be recalibrated against international standards such as USGS42 and USGS43, which unfortunately do not cover the whole range of values especially with regard to sulphur (and hydrogen) isotope results.

We agree with the reviewer, and we agree that adding this cross-lab check in response to their suggestion has improved the quality of the study. We also concur that the development of broadly available substrate-specific RMs is the way forward. Developing broadly available RMs with CNS isotope values spanning a broad range of isotopic values for keratin, chitin and other organic substrates would be important. The community should come together to develop these standards.

Line 166: RMs = reference materials

We specified this abbreviation. 

Line 253: I assume that δ13C, δ15N and δ34S values of the UBCs hair were analysed at the UCD, because the values agree to those in Ammer’s thesis.

Yes thank you for noting this. We mention this line 26-263.

Figure 1 A and B. Please explain the abbreviation “hair-obs”.

We added an explanation for this abbreviation.

Line 345: Pease add the references for the published δ34S data (also in Fig. 2).

We added these references in the figure caption.

Line 472: the lowest δ2H value was 6 months PTD

We modified this as suggested.

Line 506 f: please add that also specific geological conditions give rise to distinct baseline d34S values, in particular for Mexico the occurrence of volcanoes, leading to low δ34S values (see thesis of Ammer 2020). Even if this is not clearly visible on the maps due to the low resolution, this should be mentioned in that section.

We added this point as suggested and cited Ammer 2020.

Line 545 ff.: Not only the δ2H values of drinks, but also the δ2H values of the food most probably affect the hair δ2H values (according to the equation in Fig. 2 A). From that it can be concluded that most of the hydrogen in keratin is not from the drinking water.

We modified the sentence to more explicitly point to the influence of food on hair H isotope ratios.

Line 553: change ref 20 to ref 15.

We changed as suggested.

Line 584 f: add: ... where the individual grew up and lived..., and better “the places of residence earlier in the individuals’ life” (because the place of birth cannot be identified)

We changed as suggested.

Line 588 ff: For description of the living phases of Mr. Halifax I’d suggest to mention first the former and then the more recent months of life, e.g. during the 15 to 5 month PTD interval. The shift of the isotope values along the hair strand as a result of dietary changes is always from the older to the younger part of the hair. Consequently, the shift to higher δ13C values of ~-18 ‰ was between 12 and 10 months PTD, followed in turn by a decrease of the δ13C values between 10 and 5 months PTD (from that you may conclude that he possibly entered eastern US or eastern Canada about 12 months PTD). Low δ13C values (~-19 ‰) during the last 5 months PTD are compatible...

We rewrote this paragraph as suggested.

Line 607: For δ15N values in the hair strand I’d suggest to write: .... appear to record a shift from higher to lower δ15N values around 5 months PTD.

We modified this sentence as suggested.

Line 631 ff: ...based on the results of δ2H and δ34S analyses of bulk hair samples taken from deceased UBCs found at the US-Mexico border could help to identify...

We modified as suggested.

Line 639: Screening is also possible for current unknown deceased, not only for cold-cases.

We added this sentence as suggested.

Please note that based on the Iupac guidelines and according to Coplen 2011, “δ” should be written in italic font throughout the manuscript.

• Coplen, T. B. (2011). Guidelines and recommended terms for expression of stable‐isotope‐ratio and gas‐ratio measurement results. Rapid communications in mass spectrometry, 25(17), 2538-2560

We rewrote all delta in italics throughout the manuscript.

Reviewer #2: This manuscript is much improved over the first version. I believe the authors have addressed all the reviewer comments. I only had very minor technical edits in the attached PDF.

The pdf provided to us after re-review was identical to the pdf provided in the first round of review, and we have already addressed the comments it contains. After enquiring with the editorial staff we were informed that no new marked document was available. 

7. PLOS authors have the option to publish the peer review history of their article (what does this mean?). If published, this will include your full peer review and any attached files.

Do you want your identity to be public for this peer review? For information about this choice, including consent withdrawal, please see our Privacy Policy.

Reviewer #1: No

Reviewer #2: No

---

## [Editor Report · Decision Letter 2]

26 Sep 2022

Multi-isotopes in Human Hair: A tool to initiate Cross-border Collaboration in International Cold-Cases

PONE-D-22-13353R2

Dear Dr. Bataille,

We’re pleased to inform you that your manuscript has been judged scientifically suitable for publication and will be formally accepted for publication once it meets all outstanding technical requirements.

Kind regards,

Fabio Marzaioli, Ph.D

Academic Editor

PLOS ONE

---

## [Editor Report · Acceptance letter]

29 Sep 2022

PONE-D-22-13353R2 

Multi-isotopes in Human Hair: A tool to initiate Cross-border Collaboration in International Cold-Cases 

Dear Dr. Bataille:

I'm pleased to inform you that your manuscript has been deemed suitable for publication in PLOS ONE. Congratulations! Your manuscript is now with our production department. 

Kind regards, 

on behalf of

Dr. Fabio Marzaioli 

Academic Editor

PLOS ONE